# TOWARDS DEMYSTIFYING REPRESENTATION LEARNING WITH NON-CONTRASTIVE SELF-SUPERVISION

## ABSTRACT

Non-contrastive methods of self-supervised learning (such as BYOL and SimSiam) learn representations by minimizing the distance between two views of the same image. These approaches have achieved remarkable performance in practice, but it is not well understood how the representation is learned based on the augmentation process. Tian et al. (2021) explained why the representation does not collapse to zero and proposed DirectPred that sets the predictor directly. In our work, we analyze a generalized version of DirectPred, called **DirectSet**($\alpha$). We show that in a simple linear network, DirectSet($\alpha$) provably learns a desirable projection matrix and also reduces the sample complexity on downstream tasks. Our analysis suggests that weight decay acts as an implicit threshold that discard the features with high variance under augmentation, and keep the features with low variance. Inspired by our theory, we simplify DirectPred by removing the expensive eigen-decomposition step. On CIFAR-10, CIFAR-100, STL-10 and ImageNet, **DirectCopy**, our simpler and more computationally efficient algorithm, rivals or even outperforms DirectPred.

## 1 INTRODUCTION

Self-supervised learning recently emerges as a promising direction to learn representations without manual labels. While contrastive learning (Oord et al., 2018; Tian et al., 2019; Bachman et al., 2019; He et al., 2020; Chen et al., 2020a) minimizes the distance of representation between positive pairs, and maximizes such distances between negative pairs, recently, *non-contrastive* self-supervised learning (abbreviated as **nc-SSL**) is able to learn nontrivial representation with only positive pairs, using an extra predictor and a stop-gradient operation. Furthermore, the learned representation shows comparable (or even better) performance for downstream tasks (e.g., image classification) (Grill et al., 2020; Chen & He, 2020). This brings about two fundamental questions: (1) why the learned representation does not collapse to trivial (i.e., constant) solutions, and (2) without negative pairs, what representation nc-SSL learns from the training and how the learned representation reduces the sample complexity in downstream tasks.

While many theoretical results on contrastive SSL (Arora et al., 2019; Lee et al., 2020; Tosh et al., 2020; Wen & Li, 2021) do exist, similar study on nc-SSL has been very rare. As one of the first work towards this direction, Tian et al. (2021) show that while the global optimum of the non-contrastive loss is indeed a trivial one, following gradient direction in nc-SSL, one can find a *local* optimum that admits a nontrivial representation. Based on their theoretical findings on gradient-based methods, they proposed a new approach, DirectPred, that directly sets the predictor using the eigen-decomposition of the correlation matrix of input before the predictor, rather than updating it with gradient methods. As a method for nc-SSL, DirectPred shows comparable or better performance in multiple datasets, including CIFAR-10 (Krizhevsky et al., 2009), STL-10 (Coates et al., 2011) and ImageNet (Deng et al., 2009), compared to BYOL (Grill et al., 2020) and SimSiam (Chen & He, 2020) that optimize the predictor using gradient descent.

While Tian et al. (2021) address the first question, i.e., why the learned representation does not collapse, they do not address the second question, i.e., what representation is learned in nc-SSL and how the learned representation is related to the data distribution and augmentation process and in turn whether it reduces the sample complexity in downstream tasks.

**Main Contributions.**   In this paper, we make a first attempt towards the second question, by studying a family of algorithms named DirectSet($\alpha$), in which the DirectPred algorithm proposed by Tian et al. (2021) is a special case with $\alpha = 1/2$. Our contribution is two-folds:

First, we perform a theoretical analysis on DirectSet($\alpha$) with linear networks. Our analysis shows that there exists an *implicit threshold*, determined by weight decay parameter $\eta$, that governs which features are learned and which are discarded. More specifically, the threshold is applied to the variance of the feature across different data augmentations (or "views") of the same instance: *nuisance features* (features with high variances under augmentation) are discarded, while *invariant features* (i.e., with low variances) are kept. We further make a formal statement on the sample complexity of the learning process and performance guarantees of the downstream tasks, in the linear setting similar to Tian et al. (2021). To our knowledge, this is the first sample complexity result in nc-SSL.

Second, we show that *DirectCopy*, a special case of DirectSet($\alpha$) when $\alpha = 1$, performs comparably with (or even outperforms) DirectPred in downstream tasks in CIFAR-10, CIFAR-100, STL-10 and ImageNet. In DirectCopy, the predictor can be set *without* the expensive eigen-decomposition operation, which makes DirectCopy much simpler and more efficient than DirectPred.

**Related works.**   In nc-SSL, different techniques are proposed to avoid collapsing. BYOL and SimSiam use an extra predictor and stop gradient operation. Beyond these, BatchNorm (including its variants (Richemond et al., 2020)), de-correlation (Zbontar et al., 2021; Bardes et al., 2021; Hua et al., 2021), whitening (Ermolov et al., 2021), centering (Caron et al., 2021), and online clustering (Caron et al., 2020) are all effective ways to enforce implicit contrastive constraints among samples for collapsing prevention. We study BYOL and SimSiam as representative nc-SSL methods.

**Organization.**   The paper is organized as follows. Section 2-3 introduce DirectSet($\alpha$), prove it learns a projection matrix onto the invariant features, and the learned representation reduces sample complexity in downstream tasks. Section 4 demonstrates that DirectCopy achieves comparable or even better performance than the original DirectPred algorithm in various datasets, and Section 5 shows ablation experiments. Finally, limitation and future works are discussed in Section 6-7.

## 2   PRELIMINARIES

### 2.1   NOTATIONS

We use $I_d$ to denote the $d \times d$ identity matrix and simply write $I$ when the dimension is clear. For any linear subspace $S$ in $\mathbb{R}^d$, we use $P_S \in \mathbb{R}^{d \times d}$ to denote the projection matrix on $S$. More precisely, the projection matrix $P_S$ equals $UU^\top$, where the columns of $U$ constitute a set of orthonormal bases for subspace $S$. We use $\mathcal{N}(\mu, \Sigma)$ to denote the Gaussian distribution with mean $\mu$ and covariance $\Sigma$.

We use $\|\cdot\|$ to denote spectral norm for a matrix, or $\ell_2$ norm for a vector and use $\|\cdot\|_F$ to denote Frobenius norm for a matrix. For a real symmetric matrix $A \in \mathbb{R}^{d \times d}$ whose eigen-decomposition is $\sum_{i=1}^d \lambda_i u_i u_i^\top$, we use $|A|$ to denote $\sum_{i=1}^d |\lambda_i| u_i u_i^\top$. If $A$ is also positive semi-definite, we use $A^\alpha$ to denote $\sum_{i=1}^d \lambda_i^\alpha u_i u_i^\top$ for any positive $\alpha \in \mathbb{R}$.

### 2.2   DIRECTSET($\alpha$) AND DIRECTCOPY

In nc-SSL, recent methods as BYOL (Grill et al., 2020) and SimSiam (Chen & He, 2020) employ a dual pair of Siamese networks (Bromley et al., 1994): one side is a composition of an online network (including a projector) and a predictor network, the other side is a target network (see Figure 1 for a simple example). The target network has the same architecture as the online network, but has potentially different weights. Given an input $x$, two augmented views $x_1, x_2$ are generated, and the network is trained to match the representation of $x_1$ (through the online network and the predictor network) and the representation of $x_2$ (through the target network). More precisely, suppose the online network and the target network are two mappings $f_\theta, f_{\theta_a} : \mathbb{R}^d \mapsto \mathbb{R}^h$ and the predictor network is a mapping $g_{\theta_p} : \mathbb{R}^h \mapsto \mathbb{R}^h$, the network is trained to minimize the following loss:

$$L(\theta, \theta_p, \theta_a) := \frac{1}{2} \mathbb{E}_{x_1, x_2} \left\| \frac{g_{\theta_p}(f_\theta(x_1))}{\|g_{\theta_p}(f_\theta(x_1))\|} - \text{StopGrad}\left( \frac{f_{\theta_a}(x_2)}{\|f_{\theta_a}(x_2)\|} \right) \right\|^2.$$

In BYOL and SimSiam, the online network and the target network are trained by running gradient methods on $L$. The target network is not trained by gradient methods; instead, it is directly set with

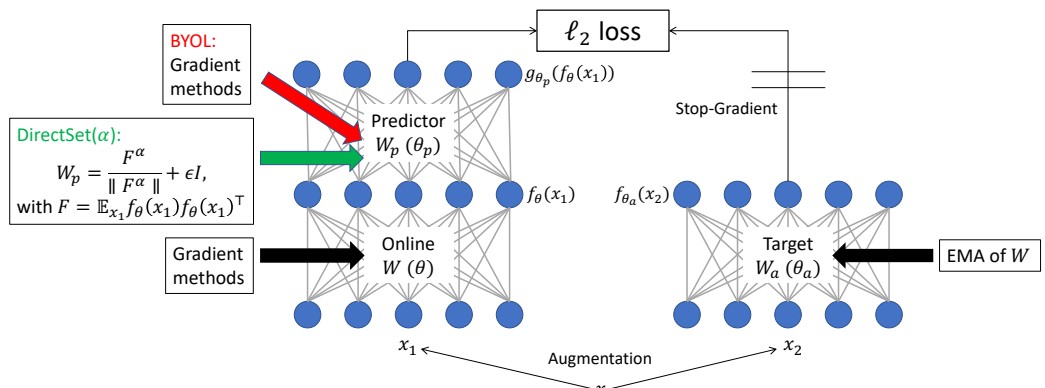

Figure 1: Problem Setup. Comparison between BYOL and DirectSet($\alpha$) on a linear network.

the weights in the online network (Chen & He, 2020) or an exponential moving average (EMA) of the online network (Grill et al., 2020; He et al., 2020; Chen et al., 2020b).

The DirectSet($\alpha$) algorithm, as shown in Figure 1, directly sets the predictor based on the correlation matrix $F$ of the predictor inputs:

$$W_p = \frac{F^\alpha}{\|F^\alpha\|} + \epsilon I,$$

where $F = \mathbb{E}_{x_1} f_\theta(x_1) f_\theta(x_1)^\top$. In practice, $F$ is estimated by a moving average over batches. That is, $\hat{F} = \mu \hat{F} + (1 - \mu)\mathbb{E}_B[f_\theta(x_1)f_\theta(x_1)^\top]$, where $\mathbb{E}_B$ is the expectation over one batch.

In the original DirectPred proposed by Tian et al. (2021), $\alpha$ is fixed at $1/2$. To compute $\hat{F}^{1/2}$, one needs to first compute the eigen-decomposition of $\hat{F}$, and then taking the root of each eigenvalue. This step of eigen-decomposition can be expensive especially when the representation dimension $h$ is high. To avoid the eigen-decomposition step, we propose DirectCopy ($\alpha = 1$), in which the predictor $W_p$ is a direct copy of the $\hat{F}$ (with normalization and regularization)[1]. As we shall see, DirectCopy enjoys both theoretical guarantees and strong empirical performance.

## 3 THEORETICAL ANALYSIS OF DIRECTSET($\alpha$)

Deep linear networks have been widely used as a tractable theoretical model for studying nonconvex loss landscapes (Kawaguchi, 2016; Du & Hu, 2019; Laurent & Brecht, 2018) and nonlinear learning dynamics (Saxe et al., 2013; 2019; Lampinen & Ganguli, 2018; Arora et al., 2018). However, most of them are for supervised learning setting. Tian et al. (2021) analyzed nc-SSL on a linear network, but did not analyze their proposed approach DirectPred. Here, we analyze the representation learning process of DirectSet($\alpha$) on a minimal setting where the online network $f_\theta$ is a single linear layer. We also verify DirectSet($\alpha$) works for practical nonlinear deep models and realistic datasets.

### 3.1 SETUP

In this subsection, we define the network model, data distribution and simplify DirectSet($\alpha$) algorithm for our theoretical analysis. We consider the following network model (see Figure 1),

**Assumption 1** (Linear network model). *The online, predictor and target network are all single-layer linear network without bias, with weight matrices denoted as $W, W_p, W_a \in \mathbb{R}^{d \times d}$ respectively.*

For the data distribution, we assume the input space is a direct sum of a invariant feature subspace and a nuisance feature subspace. Specifically, we assume

**Assumption 2** (Data distribution). *The input $x$ is sampled from $\mathcal{N}(0, I_d)$, and its augmented view $x_1, x_2$ are independently sampled from $\mathcal{N}(x, \sigma^2 P_B)$, where $B$ is a $(d - r)$-dimensional subspace. We denote $S$ as the orthogonal subspace of $B$ in $\mathbb{R}^d$.*

---

[1]Computing the spectral norm of $\hat{F}$ is much faster than computing the eigen-decomposition of $\hat{F}$, because the former only needs the top eigen-vector of $\hat{F}$. Table 4 shows that the spectral norm can also be replaced by Frobenius norm or no normalization, and similar performance can be achieved.

In this simple data distribution, subspace $S$ corresponds to the features that are invariant to augmentations and its orthogonal subspace $B$ is the nuisance subspace which the augmentation changes. We will prove that DirectSet($\alpha$) can learn the projection matrix onto $S$ subspace. Note in the previous work (Tian et al., 2021), they assumed the covariance of the augmentation distribution to be $\sigma^2 I$ and did not study what representation is learned.

For the convenience of analysis, we consider a simplified version of DirectSet($\alpha$). We compute the loss function without normalizing the two representations, so the population loss is

$$L(W, W_a, W_p) := \frac{1}{2} \mathbb{E}_{x_1, x_2} \left\| W_p W x_1 - \text{StopGrad}\left(W_a x_2\right) \right\|^2, \tag{1}$$

and the empirical loss is

$$\hat{L}(W, W_p, W_a) := \frac{1}{2n} \sum_{i=1}^{n} \left\| W_p W x_1^{(i)} - \text{StopGrad}(W_a x_2^{(i)}) \right\|^2, \tag{2}$$

where $x^{(i)}$'s are independently sampled from $\mathcal{N}(0, I)$, and augmented views $x_1^{(i)}$ and $x_2^{(i)}$ are independently sampled from $\mathcal{N}(x^{(i)}, \sigma^2 P_B)$. To train our model, first, we initialize $W$ as $\delta I$ with $\delta$ a positive real number. We run gradient flow or gradient descent on online network $W$ with weight decay $\eta$, and set the the target network $W_a = W$. For clarity of presentation, when training on the population loss, we set $W_p$ as $(W \mathbb{E}_x xx^\top W^\top)^\alpha = (WW^\top)^\alpha$ instead of $(W \mathbb{E}_{x_1} x_1 x_1^\top W^\top)^\alpha$ as in practice; when training on the empirical loss, we set $W_p$ as $(W \frac{1}{n} \sum_{i=1}^{n} x^{(i)} [x^{(i)}]^\top W^\top)^\alpha$. Here, we set the predictor regularization $\epsilon = 0$ and its influence will be studied in Section 5.

In the following, DirectSet($\alpha$) is shown to recover the projection matrix $P_S$ with polynomial number of samples. Furthermore, given that the learned matrix is close to $P_S$, the sample complexity on downstream tasks is reduced.

## 3.2 Gradient Flow on Population Loss

In this section, we show that DirectSet($\alpha$) running on the population loss with infinitesimal learning rate and $\eta$ weight decay can learn the projection matrix onto the invariant feature subspace $S$.

**Theorem 1.** *Suppose network architecture and data distribution follow Assumption 1 and Assumption 2, respectively. Suppose we initialize online network $W$ as $\delta I$, and run DirectSet($\alpha$) on population loss (see Eqn. 1) with infinitesimal step size and $\eta$ weight decay. If we set the weight decay coefficient $\eta \in \left(\frac{1}{4(1+\sigma^2)}, \frac{1}{4}\right)$ and initialization scale $\delta > \left(\frac{1 - \sqrt{1-4\eta}}{2}\right)^{1/(2\alpha)}$, then $W$ converges to $\left(\frac{1 + \sqrt{1-4\eta}}{2}\right)^{1/(2\alpha)} P_S$ when time goes to infinity.*

Theorem 1 shows that when the weight decay is in certain range, and when the initialization is large enough, the online network can converge to the desired projection matrix $P_S$ [2]. In sequel, we explain how the dynamics of $W$ leads to a projection matrix and how the weight decay and initialization scale come into play. We leave the full proof in Appendix B.1. We also consider the setting when $W_p$ is set as $(W \mathbb{E}_{x_1} x_1 x_1^\top W^\top)^\alpha$ in Appendix B.4 and extend the result to deep linear networks in Appendix C.

Due to the identity initialization, we can ensure that $W$ is always a real symmetric matrix and is simultaneously diagonalizable with $P_B$. We can then analyze the evolution of each eigenvalue in $W$ separately. Under our assumptions, it turns out that all the eigenvalues whose eigenvectors lie in the $B$ subspace share the same value $\lambda_B$, and all the eigenvalues in the $S$ subspace share the value $\lambda_S$ as shown in the following time dynamics:

$$\dot{\lambda}_B = \lambda_B \left[ -(1+\sigma^2) |\lambda_B|^{4\alpha} + |\lambda_B|^{2\alpha} - \eta \right], \quad \dot{\lambda}_S = \lambda_S \left[ -|\lambda_S|^{4\alpha} + |\lambda_S|^{2\alpha} - \eta \right]. \tag{3}$$

Next, we show $\lambda_B$ converges to zero and $\lambda_S$ converges to a positive number, which immediately implies that $W$ converges to some scaling of $P_S$.

---

[2] Note that Theorem 1 also holds with negative initialization $\delta < -\left(\frac{1 - \sqrt{1-4\eta}}{2}\right)^{1/(2\alpha)}$, in which case $W$ converges to $-\left(\frac{1 + \sqrt{1-4\eta}}{2}\right)^{1/(2\alpha)} P_S$. Our other results can be extended to negative $\delta$ in a similar way.

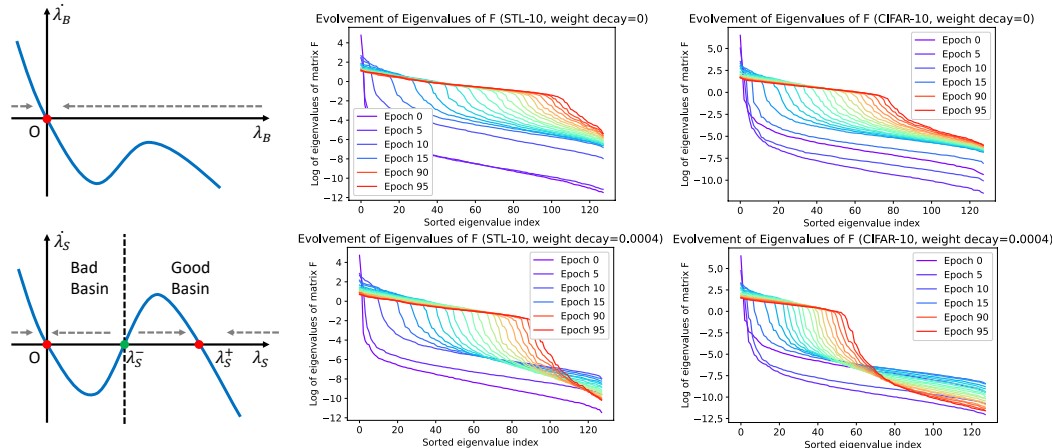

**Figure 2: Left:** With appropriate weight decay, $\lambda_B$ always converge to zero; $\lambda_S$ converges to zero when it's initialized in the bad basin and converges to positive $\lambda_S^+$ when it's initialized in the good basin. **Middle:** The evolvement of the eigenvalues of $F$ when it's trained by DirectCopy with $\epsilon = 0.2$ on STL-10. With weight decay $\eta = 0.0004$ (bottom), the eigen-spectrum at epoch 95 has sharp drop; while the drop is much milder when $\eta = 0$ (top). **Right:** Similar phenomenon on CIFAR-10 with $\epsilon = 0.3$.

Similar as the analysis in Tian et al. (2021), when $\eta > \frac{1}{4(1+\sigma^2)}$, we know $\dot{\lambda}_B < 0$ for any $\lambda_B > 0$ and $\lambda_B = 0$ is a stable stationary point, as illustrated in Figure 2 (top, left). Therefore, as long as $\eta > \frac{1}{4(1+\sigma^2)}$, $\lambda_B$ must converge to zero. When $0 < \eta < \frac{1}{4}$, there are three non-negative solutions to $\dot{\lambda}_S = 0$, which are $0, \lambda_S^- = \left(\frac{1-\sqrt{1-4\eta}}{2}\right)^{1/(2\alpha)}$ and $\lambda_S^+ = \left(\frac{1+\sqrt{1-4\eta}}{2}\right)^{1/(2\alpha)}$. As illustrated in Figure 2 (bottom, left), if initialization $\delta > \lambda_S^-$ (good basin), $\lambda_S$ converges to a positive value $\lambda_S^+$; if $0 < \delta < \lambda_S^-$ (bad basin), $\lambda_S$ converges to zero.

**Thresholding role of weight decay in feature learning:** While Tian et al. (2021) shows why nc-SSL does not collapse, one key question is how nc-SSL learns useful features and how the method determines which feature is learned. Now it is clear: the weight decay factor $\eta$ makes a call on what features should be learned. *Nuisance features* subject to significant change under data augmentation has larger variance $\sigma^2$ and $\frac{1}{4(1+\sigma^2)} < \eta$, the eigenspace corresponds to this feature goes to zero; on the other hand, *invariant features* that are robust to data augmentation has much smaller $\sigma^2$ and $\frac{1}{4(1+\sigma^2)} > \eta$ and these features are kept. In our above analysis, $B$ subspace corresponds to the nuisance features and collapses to zero; $S$ subspace corresponds to the invariant features (whose variance was assumed as zero for simplicity) and is kept after training.

Figure 2 (middle and right) shows the spectrum of $F$ (which is the correlation matrix of the predictor inputs) when the network is trained by DirectSet(1) with and without weight decay $\eta$ on STL10 and CIFAR10: when $\eta = 0$, the eigen-spectrum of $F$ in later epochs does not have a sharp drop compared with the case of $\eta = 0.0004$. This means that the nuisance features are not significantly suppressed when $\eta = 0$.

Therefore, it is crucially important to choose weight decay appropriately: a too small $\eta$ may not be sufficient to suppress the nuisance features; a too large $\eta$ can also collapse the invariant features. As shown in Section 5, both cases lead to worse downstream performance.

### 3.3 GRADIENT DESCENT ON EMPIRICAL LOSS

In this section, we then proceed to prove that DirectCopy (one special case of DirectSet($\alpha$) with $\alpha = 1$) successfully learns the projection matrix given polynomial number of samples.

**Theorem 2.** *Suppose network architecture and data distribution are as defined in Assumption 1 and Assumption 2, respectively. Suppose we initialize online network as $\delta I$, and run DirectCopy on empirical loss (see Eqn. 2) with $\gamma$ step size and $\eta$ weight decay. Suppose the noise scale $\sigma^2$ is a positive constant, the weight decay coefficient $\eta \in \left(\frac{1+\sigma^2/4}{4(1+\sigma^2)}, \frac{1+3\sigma^2/4}{4(1+\sigma^2)}\right)$ and the initialization*

scale $\delta$ is a constant at least $1/\sqrt{2}$. Choose the step size $\gamma$ as a small enough constant. For any accuracy $\hat{\epsilon} > 0$, given $n \geq poly(d, 1/\hat{\epsilon})$ number of samples, with probability at least $0.99$ there exists $t = O(\log(1/\hat{\epsilon}))$ such that (here $\widetilde{W}_t$ is the online network weights at the $t$-th step):

$$\left\| \widetilde{W}_t - \sqrt{\frac{1 + \sqrt{1 - 4\eta}}{2}} P_S \right\| \leq \hat{\epsilon}.$$

The proof proceeds by first proving that gradient descent on the population loss converges in linear rate and then couples the gradient descent dynamics on empirical loss and that on population loss. See the detailed proof in Appendix B.2.

### 3.4 SAMPLE COMPLEXITY ON DOWNSTREAM TASKS

In this section, we show that the learned representations can indeed reduce the sample complexity on the downstream tasks. We consider the following data distribution for the down-stream task:

**Assumption 3** (Downstream data distribution). *Each input $x^{(i)}$ is sampled from $\mathcal{N}(0, I_d)$ and its label $y^{(i)} = \langle x^{(i)}, w^* \rangle + \xi^{(i)}$, where $w^*$ is the ground truth vector with unit $\ell_2$ norm and $\xi^{(i)}$ is independently sampled from $\mathcal{N}(0, \beta^2)$. We assume the ground truth $w^*$ lies on an $r$-dimensional subspace $S$ and we denote the projection matrix on subspace $S$ simply as $P$.*

In practice, usually the semantically relevant features ($S$ subspace here) are invariant to augmentations and the nuisance features (orthogonal subspace of $S$) have high variance under augmentations. Therefore, by previous analysis, we expect DirectSet($\alpha$) to learn the projection matrix $P$.

Suppose $\{(x^{(i)}, y^{(i)})\}_{i=1}^n$ are $n$ training samples. Each input $x^{(i)}$ is transformed by a matrix $\hat{P} \in \mathbb{R}^{d \times d}$ (for example the learned online network $W$) to get its representation $\hat{P}x^{(i)}$. The regularized loss is then defined as $\hat{L}(w) := \frac{1}{2n} \sum_{i=1}^n \left\| \langle \hat{P}x^{(i)}, w \rangle - y^{(i)} \right\|^2 + \frac{\rho}{2} \|w\|^2$. In the below theorem, we show that when $\left\| P - \hat{P} \right\|_F$ is small, the above ridge regression can recover the ground truth $w^*$ given only $O(r)$ number of samples.

**Theorem 3.** *Suppose the downstream data distribution is as defined in Assumption 3. Suppose $\left\| \hat{P} - P \right\|_F \leq \hat{\epsilon}$ with $\hat{\epsilon} < 1$. Choose the regularizer coefficient $\rho = \hat{\epsilon}^{1/3}$. For any $\zeta < 1/2$, given $n \geq O(r + \log(1/\zeta))$ number of samples, with probability at least $1 - \zeta$, the training loss minimizer $\hat{w}$ satisfies*

$$\left\| \hat{P}\hat{w} - w^* \right\| \leq O\left( \hat{\epsilon}^{1/3} + \beta \frac{\sqrt{r} + \sqrt{\log(1/\zeta)}}{\sqrt{n}} \right).$$

In the above theorem, when $n$ is at least $O\left( \frac{\beta^2(r + \log(1/\zeta))}{\hat{\epsilon}^{2/3}} \right)$, we have $\left\| \hat{P}\hat{w} - w^* \right\| \leq O(\hat{\epsilon}^{1/3})$. Note that if we directly estimate $\hat{w}$ without transforming the inputs by $\hat{P}$, we need $\Omega(d)$ number of samples to ensure that $\|\hat{w} - w^*\| \leq o(1)$ (Wainwright, 2019). The proof of Theorem 3 follows from bounding the difference between $\hat{P}\hat{w}$ and $w^*$ by matrix concentration inequalities and matrix perturbation bounds. The full proof is in Appendix B.3.

## 4 EMPIRICAL PERFORMANCE OF DIRECTCOPY

In the previous analysis, we show DirectSet($\alpha$), and in particular DirectCopy (DirectSet($\alpha$) with $\alpha = 1$), could recover the input feature structure with polynomial samples and make the downstream task more sample efficient in a simple linear setting. Compared with the original DirectPred (DirectSet($\alpha$) with $\alpha = 1/2$), DirectCopy is a simpler and computationally more efficient algorithm since it directly set the predictor as the correlation matrix $F$, without the eigen-decomposition step. By our analysis in Theorem 1, DirectCopy also learns the projection matrix $P_S$ with larger scale [3]

---

[3]Recall that in Theorem 1 under DirectSet($\alpha$), online matrix $W$ converges to $\left( \frac{1 + \sqrt{1 - 4\eta}}{2} \right)^{1/(2\alpha)} P_S$. So with a larger $\alpha$, the scalar in front of $P_S$ becomes larger.

compared with DirectPred, which suggests that the invariant features learned by DirectCopy are stronger and more distinguishable. Next, we show that DirectCopy is on par with (or even outperforms) the original DirectPred in various datasets, when coupling with deep nonlinear models on real datasets.

## 4.1 RESULTS ON STL-10, CIFAR-10 AND CIFAR-100

We use ResNet-18 (He et al., 2016) as the backbone network, a two-layer nonlinear MLP as the projector, and a linear predictor. Unless specified otherwise, SGD is used as the optimizer with weight decay $\eta = 0.0004$. To evaluate the quality of the pre-trained representations, we follow the linear evaluation protocol. Each setting is repeated 5 times to compute the mean and standard deviation. The accuracy is reported as "mean±std". Unless explicitly specified, we use learning rate $\gamma = 0.01$, regularization $\epsilon = 0.2$ on STL-10; $\gamma = 0.02, \epsilon = 0.3$ on CIFAR-10 and $\gamma = 0.03, \epsilon = 0.3$ on CIFAR-100. See more detailed experiment settings in Appendix A.

| | Num of epochs | | |
| --- | --- | --- | --- |
| | 100 | 300 | 500 |
| *STL-10* | | | |
| DirectCopy | 77.83±0.56 | **82.01±0.28** | **82.95±0.29** |
| DirectPred | **77.86±0.16** | 78.77±0.97 | 78.86±1.15 |
| DirectPred (freq=5) | 77.54±0.11 | 79.90±0.66 | 80.28±0.62 |
| SGD baseline | 75.06±0.52 | 75.25±0.74 | 75.25±0.74 |
| *CIFAR-10* | | | |
| DirectCopy | 84.02±0.37 | **89.17±0.12** | **89.62±0.10** |
| DirectPred | **85.21±0.23** | 88.88±0.15 | 89.52±0.04 |
| DirectPred (freq=5) | 84.93±0.29 | 88.83±0.10 | 89.56±0.13 |
| SGD baseline | 84.49±0.20 | 88.57±0.15 | 89.33±0.27 |
| *CIFAR-100* | | | |
| DirectCopy | 55.40±0.19 | 61.06±0.14 | 62.23±0.06 |
| DirectPred | **56.60±0.27** | 61.65±0.18 | 62.68±0.35 |
| DirectPred (freq=5) | 56.43±0.21 | **62.01±0.22** | **63.15±0.27** |
| SGD baseline | 54.94±0.50 | 60.88±0.59 | 61.42±0.89 |

Table 1: STL-10/CIFAR-10/CIFAR-100 Top-1 accuracy of DirectCopy. The numbers for DirectPred, DirectPred (freq=5) and SGD baseline on STL-10/CIFAR-10 are obtained from Tian et al. (2021).

| | epochs |
| --- | --- |
| | 100 |
| *ImageNet* | |
| DirectCopy | **68.8** |
| DirectPred | 68.5 |
| SGD Baseline | 68.6 |

Table 2: ImageNet Top-1 accuracy of DirectCopy, DirectPred and BYOL baseline.

**STL-10:** We evaluate the quality of the learned representation after each epoch, and report the best accuracy in the first 100/300/500 epochs in Table 1. DirectCopy achieves substantially better performance than the original DirectPred and SGD baseline, especially when trained with longer epochs. DirectPred (freq=5) means the predictor is set by DirectPred every 5 batches, and is trained with gradient updates in other batches, which outperforms DirectPred in later epochs, but is still much worse than DirectCopy. The SGD baseline is obtained by training the linear predictor using SGD.

**CIFAR-10/100:** For CIFAR-10, DirectCopy is slighly worse than DirectPred at epoch 100, but catches up and gets even better performance in epoch 300 and 500 (Table 1). For CIFAR-100, at earlier epochs, the performance of DirectCopy is not as good as DirectPred, but the gap gradually diminishes in later epochs. Both DirectCopy and DirectPred outperfoms the SGD baseline. DirectPred (freq=5) achieves even better performance, but at the cost of a more complicated algorithm.

## 4.2 RESULTS ON IMAGENET

Following BYOL (Grill et al., 2020), we use ResNet-50 as the backbone and a two-layer MLP as the projector. We use LARS (You et al., 2017) optimizer and trains the model for 100 epochs. See more detailed experiment settings in Appendix A.

For fairness, we compare DirectCopy to the gradient-based baseline which uses the same-sized linear predictor as ours. As shown in Table 2, at 100-epoch, this baseline achieves 68.6 top-1 accuracy, which is already significantly higher than BYOL with two-layer predictors reported in the literature (e.g., Chen & He (2020) reports 66.5 top-1 under 100-epoch training). DirectCopy using normalized $F$ accumulated with EMA $\mu = 0.99$ on the correlation matrix, regularization parameter $\epsilon = 0.01$ achieves 68.8 under the same setting, better than this strong baseline. In contrast, DirectPred (Tian et al., 2021) achieves 68.5, slightly lower than the linear baseline.

Figure 3: **Left:** Change of $\dot{\lambda}_S$ when predictor regularization $\epsilon$ increases. **Right:** Eigenvalues of $F$ when trained by DirectCopy under different $\epsilon$ on CIFAR-10 for 100 epochs.

## 5 ABLATION STUDY

In this section, we study the influence of predictor regularization $\epsilon$, normalization method, weight decay and degree $\alpha$ on the performance of DirectCopy.

**Predictor regularization:** Table 3 shows that when the predictor regularization $\epsilon$ increases, the performance of DirectCopy on STL-10 and CIFAR-10 improves at first and then deteriorates. On STL-10, DirectCopy with $\epsilon = 1$ completely fails. On CIFAR-10, although DirectCopy with $\epsilon = 1$ achieved reasonable performance at epoch 300, it's still much worse than $\epsilon = 0.3$.

To better understand the role of $\epsilon$, we analyze the simple linear setting as in Section 3.1 while setting $W_p = WW^\top + \epsilon I$. Recall that $\lambda_B$ is the eigenvalue of $W$ in $B$ subspace and $\lambda_S$ is that in $S$ subspace. When the weight decay is appropriate, $\lambda_B$ still converges to zero. On the other hand, the dynamics for $\lambda_S$ is as follows:

$$\dot{\lambda}_S = -\lambda_S \left( \lambda_S^2 + \epsilon - \frac{1 - \sqrt{1 - 4\eta}}{2} \right) \left( \lambda_S^2 + \epsilon - \frac{1 + \sqrt{1 - 4\eta}}{2} \right).$$

Increasing $\epsilon$ shifts the two positive stationary points $\lambda_S^-, \lambda_S^+$ towards zero. As illustrated in Figure 3 (left), as $\epsilon$ increases, when $\lambda_S^+$ is still positive, the good attraction basin expands, which means $\lambda_S$ can converge to a positive value from a smaller initialization; when $\lambda_S^+$ shifts to zero, $\lambda_S$ converges to zero regardless the initialization size. See the full analysis in Appendix D.

Intuitively, a reasonable $\epsilon$ can alleviate representation collapse, but a too large $\epsilon$ also encourages representation collapse. As shown in Figure 3 (right), when $\epsilon$ increases from zero, more eigenvalues of $F$ becomes large; but when $\epsilon$ exceeds 0.3, eigenvalues of $F$ begin to collapse.

**Normalization on $F$:** In our experiments, we have been normalizing $F$ by its spectral norm before adding the regularization: $W_p = F/\|F\| + \epsilon I$. It turns out that we can also normalize $F$ by its Frobenius norm or simply skip the normalization step. In Table 4, we see comparable performance from DirectCopy with Frobenius normalization or no normalization, especially when trained longer.

|  | Number of epochs | |
|---|---|---|
|  | 100 | 300 |
| *STL-10* | | |
| $\epsilon = 0$ | 76.57±0.66 | 81.19±0.39 |
| $\epsilon = 0.1$ | **78.05±0.14** | 81.60±0.15 |
| $\epsilon = 0.2$ | 77.83±0.56 | **82.01±0.28** |
| $\epsilon = 1$ | 31.10±0.80 | 31.10±0.80 |
| *CIFAR-10* | | |
| $\epsilon = 0$ | 80.53±1.14 | 86.07±0.71 |
| $\epsilon = 0.1$ | 83.97±0.25 | 88.58±0.11 |
| $\epsilon = 0.3$ | **84.02±0.37** | **89.17±0.12** |
| $\epsilon = 1$ | 57.38±11.62 | 83.15±4.24 |

Table 3: STL-10/CIFAR-10 Top-1 accuracy of DirectCopy with varying regularization $\epsilon$.

|  | Number of epochs | |
|---|---|---|
|  | 100 | 300 |
| *STL-10* | | |
| Spectral | **77.83±0.56** | 82.01±0.28 |
| Frobenius | 77.71±0.18 | **82.06±0.28** |
| None | 77.81±0.20 | 82.00±1.24 |
| *CIFAR-10* | | |
| Spectral | 84.02±0.37 | 89.17±0.12 |
| Frobenius | **84.33±0.25** | **89.62±0.14** |
| None | 81.76±0.34 | 89.21±0.17 |

Table 4: STL-10/CIFAR-10 Top-1 accuracy of DirectCopy with $F$ matrix normalized by spectral norm/Frobenius norm or no normalization.

**Weight decay:** Table 5 shows that when weight decay $\eta$ increases, the performance of DirectCopy improves at first and then deteriorates. This fits our analysis on simple linear networks. Basically, when the weight decay $\eta$ increases, it can suppress the nuisance features more effectively, but a too large weight decay also collapses the useful features.

|  | Number of epochs | |
|---|---|---|
|  | 100 | 300 |
| *STL-10* | | |
| $\eta = 0$ | 71.94±0.93 | 78.53±0.40 |
| $\eta = 0.0004$ | **77.83±0.56** | **82.01±0.28** |
| $\eta = 0.001$ | 77.65±0.16 | 80.28±0.16 |
| $\eta = 0.01$ | 58.12±0.94 | 58.53±0.76 |
| *CIFAR-10* | | |
| $\eta = 0$ | 79.15±0.08 | 85.35±0.31 |
| $\eta = 0.0004$ | **84.02±0.37** | **89.17±0.12** |
| $\eta = 0.001$ | 83.91±0.33 | 87.75±0.16 |
| $\eta = 0.01$ | 65.31±1.19 | 65.63±1.30 |

Table 5: STL-10/CIFAR-10 Top-1 accuracy of DirectCopy with varying weight decay.

|  | Number of epochs | |
|---|---|---|
|  | 100 | 300 |
| *STL-10* | | |
| $\alpha = 2$ | 76.80±0.22 | 80.90±0.18 |
| $\alpha = 1$ | **77.83±0.56** | **82.01±0.28** |
| $\alpha = 1/2$ | 77.82±0.37 | 77.83±0.37 |
| $\alpha = 1/4$ | 76.82±0.36 | 76.82±0.36 |
| *CIFAR-10* | | |
| $\alpha = 2$ | 82.96±0.56 | 88.60±0.11 |
| $\alpha = 1$ | 84.02±0.37 | **89.17±0.12** |
| $\alpha = 1/2$ | **84.88±0.21** | 88.32±0.57 |
| $\alpha = 1/4$ | 84.78±0.21 | 87.82±0.32 |

Table 6: STL-10/CIFAR-10 Top-1 accuracy of DirectSet($\alpha$) with varying degree $\alpha$.

**Predictor degree:** We compare DirectCopy against DirectSet($\alpha$) with $\alpha = 2, 1/2, 1/4$. Table 6 shows that DirectCopy outperforms other algorithms on STL-10. On CIFAR-10, DirectCopy is slightly worse at epoch 100, but catches up in later epochs.

# 6 BEYOND LINEAR MODELS: LIMITATIONS AND DISCUSSION

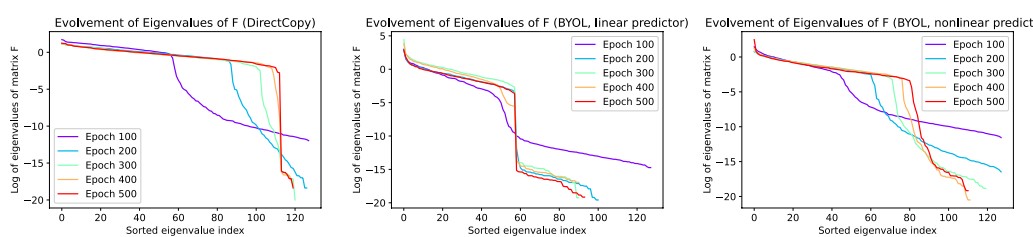

Figure 4: Eigenvalues of $F$ when trained by DirectCopy, BYOL with linear predictor and BYOL with two-layer nonlinear predictor on CIFAR-10 for different epochs. Top-1 accuracy at 500 epoch is 89.62 for Direct-Copy, 88.83 for BYOL with linear predictor and 90.25 for BYOL with two-layer nonlinear predictor.

As a linear model used to study the behavior of nc-SSL, our model does not capture all of its intriguing empirical phenomena. For example, we observed that the discarded nuisance features gradually come back after training over longer epochs. Moreover, whether it comes back or not is related to the downstream task performance. In Figure 4 on CIFAR-10 dataset, both DirectCopy and BYOL with two-layer nonlinear predictor show this resurgence of nuisance features, as well as strong performance, while BYOL with linear predictor does not seem to learn new features even when trained longer, which might explain its worse performance.

One conjecture is that at the beginning of training, weight decay prioritize the invariant features (i.e., low variance under augmentation) over nuisance ones. The invariant features then grow, building their own supporting low-level features. After that, the nuisance feature, which is also useful, are gradually picked up in later stage. Since the low-level features are already trained through previous steps of back-propagation, the nuisance features are encouraged to use them as the supporting features, rather than creating their own. In contrast, if we train both the invariant and nuisance features simultaneously, they will *compete* over the limited pool of low-level supporting features defined by the capacity of the network, leading to worse learned representations. We believe understanding these phenomena require analysis on the non-linear networks, and we leave it as future work.

# 7 CONCLUSION

In this paper, we have proved DirectSet($\alpha$) can learn the desirable projection matrix in a linear network setting and reduce the sample complexity on down-stream tasks. Our analysis sheds light on the crucial role of weight decay in nc-SSL, which discards the features that have high variance under augmentations and keep the invariant features. Inspired by the analysis, we also designed a simpler and more efficient algorithm DirectCopy, which achieves comparable or even better performance than the original DirectPred (Tian et al., 2021) on various datasets.

We view our paper as an initial step towards demystifying the representation learning in nc-SSL. Many mysteries lie beyond the explanation of the current theory and we leave them for future work.

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

## A DETAILED EXPERIMENT SETTING

**STL-10, CIFAR-10, CIFAR-100** : We use ResNet-18 (He et al., 2016) as the backbone network, a two-layer nonlinear MLP (with batch normalization, ReLU activation, hidden layer width 512, output width 128) as the projector, and a linear predictor. Unless specified otherwise, SGD is used as the optimizer with momentum 0.9, weight decay $\eta = 0.0004$ and batch size 128. The EMA parameter for the target network is set as 0.996 and the EMA parameter $\mu$ of the correlation matrix $\hat{F}$ is set as 0.5. Our code is adapted from Tian et al. (2021) [4], and we follow the same data augmentation process.

To evaluate the quality of the pre-trained representations, we follow the linear evaluation protocol. Each setting is repeated 5 times to compute the mean and standard deviation. The accuracy is reported as "mean±std". Unless explicitly specified, we use learning rate $\gamma = 0.01$, regularization $\epsilon = 0.2$ on STL-10; $\gamma = 0.02, \epsilon = 0.3$ on CIFAR-10 and $\gamma = 0.03, \epsilon = 0.3$ on CIFAR-100.

**ImageNet** : Following BYOL (Grill et al., 2020), we use ResNet-50 as the backbone and a two-layer MLP (with batch normalization, ReLU, hidden layer width 4096, output width 256) as the projector. We use LARS (You et al., 2017) optimizer and trains the model for 100 epochs, with a batch size 4096. The learning rate is 7.2, which is linearly scaled from the base learning rate 0.45 at batch size 256. Other setups such as weight decay ($\eta = 1e^{-6}$), target EMA (scheduled from 0.99 to 1), augmentation recipe (color jitters, blur, etc.), and linear evaluation protocol are the same as BYOL.

## B PROOFS OF SINGLE-LAYER LINEAR NETWORKS

### B.1 GRADIENT FLOW ON POPULATION LOSS

In this section, we give the proof of Theorem 1, which shows that DirectSet($\alpha$) running on the population loss with infinitesimal learning rate and $\eta$ weight decay can learn the projection matrix onto subspace $S$.

**Theorem 1.** *Suppose network architecture and data distribution follow Assumption 1 and Assumption 2, respectively. Suppose we initialize online network $W$ as $\delta I$, and run DirectSet($\alpha$) on population loss (see Eqn. 1) with infinitesimal step size and $\eta$ weight decay. If we set the weight decay coefficient $\eta \in \left( \frac{1}{4(1+\sigma^2)}, \frac{1}{4} \right)$ and initialization scale $\delta > \left( \frac{1-\sqrt{1-4\eta}}{2} \right)^{1/(2\alpha)}$, then $W$ converges to $\left( \frac{1+\sqrt{1-4\eta}}{2} \right)^{1/(2\alpha)} P_S$ when time goes to infinity.*

As we already mentioned in the main text, Theorem 1 is proved by analyzing each eigenvalue of $W$ separately. We show that the eigenvalues in the $B$ subspace converge to zero, and the eigenvalues in the $S$ subspace converge to the same positive number, which immediately implies that $W$ converges to a scaling of the projection matrix $P_S$.

**Proof of Theorem 1.** We can compute the gradient in terms of $W$ as follows,

$$\nabla L(W) = \mathbb{E}_{x_1, x_2} W_p^\top \left( W_p W x_1 - W_a x_2 \right) x_1^\top$$
$$= W_p^\top \left( W_p W \mathbb{E}_{x_1} x_1 x_1^\top - W_a \mathbb{E}_{x_1, x_2} x_2 x_1^\top \right).$$

Note that the two augmented views $x_1, x_2$ are sampled by first sampling input $x$ from $\mathcal{N}(0, I_d)$, and then independently sampling $x_1, x_2$ from $\mathcal{N}(x, \sigma^2 P_B)$. Therefore, we know $\mathbb{E}_{x_1} x_1 x_1^\top = I + \sigma^2 P_B$ and $\mathbb{E}_{x_1, x_2} x_2 x_1^\top = I$. Recall that we run gradient flow on $W$ with weight decay $\eta$, so the dynamics on $W$ is as follows:

$$\dot{W} = W_p^\top \left( -W_p W (I + \sigma^2 P_B) + W_a \right) - \eta W,$$

where the first term comes from the gradient and the second term is due to weight decay.

Since $W$ is initialized as $\delta I$, and $W_a = W, W_p = (WW^\top)^\alpha$, so we know initially $W, W_p, W_a, I$ and $P_B$ are all simultaneously diagonalizable, which then implies $\dot{W}$ is simultaneously diagonalizable

---
[4]Their open source code is at https://github.com/facebookresearch/luckmatters/tree/main/ssl

with $W$. This argument can continue to show that at any time point, $W, W_p, W_a, I$ and $P_B$ are all simultaneously diagonalizable. Since $W$ is always a real symmetric matrix, we have $W_p = (WW^\top)^\alpha = |W|^{2\alpha}$. The dynamics on $W$ can then be written as

$$\dot{W} = |W|^{2\alpha}\left(-|W|^{2\alpha}W(I + \sigma^2 P_B) + W\right) - \eta W$$
$$= W\left(-(I + \sigma^2 P_B)|W|^{4\alpha} + |W|^{2\alpha} - \eta\right).$$

Let the eigenvalue decomposition of $W$ be $\sum_{i=1}^d \lambda_i u_i u_i^\top$, with $\text{span}(\{u_{d-r+1}, \cdots, u_d\})$ equals to subspace $B$. We can separately analyze the dynamics of each $\lambda_i$. Furthermore, we know $\lambda_1, \cdots, \lambda_r$ have the same value $\lambda_S$ and $\lambda_{d-r+1}, \cdots, \lambda_d$ have the same value $\lambda_B$. Next, we separately show that $\lambda_B$ converge to zero and $\lambda_S$ converges to a positive value.

**Dynamics for $\lambda_B$:**  We can write down the dynamics for $\lambda_B$ as follows:

$$\dot{\lambda}_B = \lambda_B\left[-(1 + \sigma^2)|\lambda_B|^{4\alpha} + |\lambda_B|^{2\alpha} - \eta\right]$$

Similar as the analysis in Tian et al. (2021), when $\eta > \frac{1}{4(1+\sigma^2)}$, we know $\dot{\lambda}_B < 0$ for any $\lambda_B > 0$ and $\lambda_B = 0$ is a critical point. This means, as long as $\eta > \frac{1}{4(1+\sigma^2)}$, $\lambda_B$ must converge to zero.

**Dynamics for $\lambda_S$:**  We can write down the dynamics for $\lambda_S$ as follows:

$$\dot{\lambda}_S = \lambda_S\left[-|\lambda_S|^{4\alpha} + |\lambda_S|^{2\alpha} - \eta\right].$$

When $0 < \eta < \frac{1}{4}$, we know $\dot{\lambda}_S > 0$ for $\lambda_S^{2\alpha} \in \left(\frac{1-\sqrt{1-4\eta}}{2}, \frac{1+\sqrt{1-4\eta}}{2}\right)$ and $\dot{\lambda}_S < 0$ for $\lambda_S^{2\alpha} \in \left(\frac{1+\sqrt{1-4\eta}}{2}, \infty\right)$. Furthermore, we know $\dot{\lambda}_S = 0$ when $\lambda_S^{2\alpha} = \frac{1+\sqrt{1-4\eta}}{2}$. Therefore, as long as $0 < \eta < \frac{1}{4}$ and initialization $\delta^{2\alpha} > \frac{1-\sqrt{1-4\eta}}{2}$, we know $\lambda_S^{2\alpha}$ converges to $\frac{1+\sqrt{1-4\eta}}{2}$.

Overall, we know when $\frac{1}{4(1+\sigma^2)} < \eta < \frac{1}{4}$ and $\delta > \left(\frac{1-\sqrt{1-4\eta}}{2}\right)^{1/(2\alpha)}$, we have $\lambda_B$ converge to zero and $\lambda_S$ converge to $\left(\frac{1+\sqrt{1-4\eta}}{2}\right)^{1/(2\alpha)}$. That is, matrix $W$ converges to $\left(\frac{1+\sqrt{1-4\eta}}{2}\right)^{1/(2\alpha)} P_S$.

$\square$

## B.2   GRADIENT DESCENT ON EMPIRICAL LOSS

In this section, we prove that DirectCopy successfully learns the projection matrix given polynomial number of samples.

**Theorem 2.** *Suppose network architecture and data distribution are as defined in Assumption 1 and Assumption 2, respectively. Suppose we initialize online network as $\delta I$, and run DirectCopy on empirical loss (see Eqn. 2) with $\gamma$ step size and $\eta$ weight decay. Suppose the noise scale $\sigma^2$ is a positive constant, the weight decay coefficient $\eta \in \left(\frac{1+\sigma^2/4}{4(1+\sigma^2)}, \frac{1+3\sigma^2/4}{4(1+\sigma^2)}\right)$ and the initialization scale $\delta$ is a constant at least $1/\sqrt{2}$. Choose the step size $\gamma$ as a small enough constant. For any accuracy $\hat{\epsilon} > 0$, given $n \geq poly(d, 1/\hat{\epsilon})$ number of samples, with probability at least $0.99$ there exists $t = O(\log(1/\hat{\epsilon}))$ such that (here $\widetilde{W}_t$ is the online network weights at the $t$-th step):*

$$\left\|\widetilde{W}_t - \sqrt{\frac{1 + \sqrt{1 - 4\eta}}{2}} P_S\right\| \leq \hat{\epsilon}.$$

When running gradient descent on the empirical loss, the eigenspace of $\widetilde{W}_t$ can shift and become no longer simultaneously diagonalizable with $P_B$. So we cannot independently analyze each eigenvalue of $\widetilde{W}_t$ as before, which brings significant challenge into the analysis. Instead of directly analyzing the dynamics of $\widetilde{W}_t$, we first show that the gradient descent iterates $W_t$ on the population loss converges to $P_S$ in linear rate, and then show that $\widetilde{W}_t$ stays close to $W_t$ within certain iterations.

**Lemma 1.** *In the setting of Theorem 2, let $W_t$ be the gradient descent iterations on the population loss L. Given any accuracy $\hat{\epsilon} > 0$, for any $t \geq C \log(1/\hat{\epsilon})$, we have*

$$\left\| W_t - \sqrt{\frac{1 + \sqrt{1 - 4\eta}}{2}} P_S \right\| \leq \hat{\epsilon},$$

*where $C$ is a positive constant.*

The proof of Lemma 1 is similar as the gradient flow analysis in Section 3.2. Next, we show that the gradient descent trajectory on the empirical loss stays close to the gradient descent trajectory on the population loss within $O(\log(1/\hat{\epsilon}))$ iterations.

**Lemma 2.** *In the setting of Theorem 2, let $W_t$ be the gradient descent iterations on the population loss and let $\widetilde{W}_t$ be the gradient descent iterations on the empirical loss. For any accuracy $\hat{\epsilon} > 0$, given $n \geq poly(d, 1/\hat{\epsilon})$ number of samples, with probability at least $0.99$, for any $t \leq C \log(1/\hat{\epsilon})$, we have*

$$\left\| \widetilde{W}_t - W_t \right\| \leq \hat{\epsilon},$$

*where the constant $C$ comes from Lemma 1.*

Then the proof of Theorem 2 directly follows from Lemma 1 and Lemma 2.

**Proof of Theorem 2.** According to Lemma 1, we know given any accuracy $\hat{\epsilon}'$, for $t = C \log(1/\hat{\epsilon})$, we have

$$\left\| W_t - \sqrt{\frac{1 + \sqrt{1 - 4\eta}}{2}} P_S \right\| \leq \hat{\epsilon}',$$

where $C$ is a positive constant.

According to Lemma 2, we know given $n \geq poly(d, 1/\hat{\epsilon}')$ number of samples, with probability at least $0.99$,

$$\left\| \widetilde{W}_t - W_t \right\| \leq \hat{\epsilon}'.$$

Therefore, we have

$$\left\| \widetilde{W}_t - \sqrt{\frac{1 + \sqrt{1 - 4\eta}}{2}} P_S \right\| \leq \left\| W_t - \sqrt{\frac{1 + \sqrt{1 - 4\eta}}{2}} P_S \right\| + \left\| \widetilde{W}_t - W_t \right\| \leq 2\hat{\epsilon}'.$$

Replacing $\hat{\epsilon}'$ by $\hat{\epsilon}/2$ finishes the proof. $\qquad\square$

In section B.2.1, we give the proof of Lemma 1 and Lemma 2. Proofs of some technical lemmas are left in Appendix B.5.

### B.2.1 Proofs for Lemma 1 and Lemma 2

**Proof of Lemma 1.** Similar as in Theorem 1, we can show that at any step $t$, $W_t$ is simultaneously diagonalizable with $W_{a,t}, W_{p,t}, I$ and $P_B$. The update on $W_t$ is as follows,

$$W_{t+1} = W_t + \gamma W_t \left( -(I + \sigma^2 P_B) W_t^4 + W_t^2 - \eta \right).$$

Let the eigenvalue decomposition of $W_t$ be $\sum_{i=1}^d \lambda_{i,t} u_i u_i^\top$, with $span(\{u_{d-r+1}, \cdots, u_d\})$ equals to subspace $B$. We can separately analyze the dynamics of each $\lambda_{i,t}$. Furthermore, we know $\lambda_{1,t}, \cdots, \lambda_{r,t}$ have the same value $\lambda_{S,t}$ and $\lambda_{d-r+1,t}, \cdots, \lambda_{d,t}$ have the same value $\lambda_{B,t}$. Next, we separately show that $\lambda_{B,t}$ converge to zero and $\lambda_{S,t}$ converges to a positive value in linear rate.

**Dynamics of $\lambda_{B,t}$:** We show that

$$0 \leq \lambda_{B,t} \leq (1 - \gamma C_1)^t \delta$$

for any step size $\gamma \leq C_2$, where $C_1, C_2$ are two positive constants.

According to the gradient update, we have

$$\lambda_{B,t+1} = \lambda_{B,t} + \gamma\lambda_{B,t}\left[-(1+\sigma^2)\lambda_{B,t}^4 + \lambda_{B,t}^2 - \eta\right].$$

We only need to prove that for any $\lambda_{B,t} \in [0, \delta]$, we have

$$-(1+\sigma^2)\lambda_{B,t}^4 + \lambda_{B,t}^2 - \eta = -\Theta(1).$$

This is true since $\eta \in \left(\frac{1+\sigma^2/4}{4(1+\sigma^2)}, \frac{1+3\sigma^2/4}{4(1+\sigma^2)}\right)$ and $\sigma^2, \delta$ are two positive constants.

**Dynamics of $\lambda_S$:** We show that

$$0 \le \left|\lambda_{S,t}^2 - \frac{1+\sqrt{1-4\eta}}{2}\right| \le (1-\gamma C_3)^t \left|\delta^2 - \frac{1+\sqrt{1-4\eta}}{2}\right|$$

for any step size $\gamma \le C_4$, where $C_3, C_4$ are two positive constants.

There are two cases to consider: when the initialization scale $\delta^2 \in [1/2, \frac{1+\sqrt{1-4\eta}}{2}]$, we prove

$$0 \le \frac{1+\sqrt{1-4\eta}}{2} - \lambda_{B,t}^2 \le (1-\gamma C_3)^t \left(\frac{1+\sqrt{1-4\eta}}{2} - \delta^2\right);$$

when the initialization scale $\delta^2 > \frac{1+\sqrt{1-4\eta}}{2}$, we prove

$$0 \le \lambda_{B,t}^2 - \frac{1+\sqrt{1-4\eta}}{2} \le (1-\gamma C_3)^t \left(\delta^2 - \frac{1+\sqrt{1-4\eta}}{2}\right).$$

We focus on the second case; the proof for the first case is similar.

According to the gradient update, we have

$$\begin{aligned}
\lambda_{S,t+1} &= \lambda_{S,t} + \gamma\lambda_{S,t}\left[-\lambda_{S,t}^4 + \lambda_{S,t}^2 - \eta\right]\\
&= \lambda_{S,t} - \gamma\lambda_{S,t}\left(\lambda_{S,t}^2 - \frac{1-\sqrt{1-4\eta}}{2}\right)\left(\lambda_{S,t}^2 - \frac{1+\sqrt{1-4\eta}}{2}\right)
\end{aligned}$$

We only need to show that $\lambda_{S,t}\left(\lambda_{S,t}^2 - \frac{1-\sqrt{1-4\eta}}{2}\right) = \Theta(1)$ for any $\lambda_{S,t}^2 \in [\frac{1+\sqrt{1-4\eta}}{2}, \delta]$. This is true because $\eta \in \left(\frac{1+\sigma^2/4}{4(1+\sigma^2)}, \frac{1+3\sigma^2/4}{4(1+\sigma^2)}\right)$ and $\sigma^2, \delta$ are two positive constants.

Overall, we know that there exists constant step size such that after $t = O(\log(1/\hat{\epsilon}))$ steps, we have

$$0 \le \lambda_{B,t} \le \hat{\epsilon} \text{ and } \left|\lambda_{S,t} - \sqrt{\frac{1+\sqrt{1-4\eta}}{2}}\right| \le \hat{\epsilon}.$$

This then implies,

$$\left\|W_t - \sqrt{\frac{1+\sqrt{1-4\eta}}{2}}P_S\right\| \le \hat{\epsilon}.$$

$\square$

**Proof of Lemma 2.** We know the update on $\widetilde{W}_t$ is

$$\widetilde{W}_{t+1} - \widetilde{W}_t = \gamma\widetilde{W}_{p,t}^\top\left(-\widetilde{W}_{p,t}\widetilde{W}_t\left(\frac{1}{n}\sum_{i=1}^n x_1^{(i)}[x_1^{(i)}]^\top\right) + \widetilde{W}_{a,t}\left(\frac{1}{n}\sum_{i=1}^n x_1^{(i)}[x_2^{(i)}]^\top\right)\right) - \gamma\eta\widetilde{W}_t,$$

and the update on $W_t$ is

$$W_{t+1} - W_t = \gamma W_{p,t}^\top\left(-W_{p,t}W_t\left(I + \sigma^2 P_B\right) + W_{a,t}\right) - \gamma\eta W_t.$$

Next, we bound $\left\|\widetilde{W}_{t+1} - \widetilde{W}_t - (W_{t+1} - W_t)\right\|$. According to Lemma 3, we know with probability at least $1 - O(d^2)\exp\left(-\Omega(\hat{\epsilon}'^2 n/d^2)\right)$,

$$\left\|\frac{1}{n}\sum_{i=1}^n x_1^{(i)}[x_1^{(i)}]^\top - I - \sigma^2 P_B\right\|, \left\|\frac{1}{n}\sum_{i=1}^n x_1^{(i)}[x_2^{(i)}]^\top - I\right\|, \left\|\frac{1}{n}\sum_{i=1}^n x^{(i)}[x^{(i)}]^\top - I\right\| \le \hat{\epsilon}'.$$

Recall that we set $\widetilde{W}_{a,t} = \widetilde{W}_t$ and set $W_{a,t}$ as $W_t$, so we have $\left\|\widetilde{W}_{a,t} - W_{a,t}\right\| = \left\|\widetilde{W}_t - W_t\right\|$. Also since we set $\widetilde{W}_{p,t} = \widetilde{W}_t \left(\frac{1}{n}\sum_{i=1}^n x^{(i)}[x^{(i)}]^\top\right)\widetilde{W}_t^\top$ and set $W_{p,t} = W_t W_t^\top$, we have $\left\|\widetilde{W}_{p,t} - W_{p,t}\right\| = O\left(\left\|\widetilde{W}_t - W_t\right\| + \hat{\epsilon}'\right)$ since $\|W_t\| = O(1)$.

Combing the above bounds and recall $\gamma$ is a constant, we have

$$\left\|\widetilde{W}_{t+1} - \widetilde{W}_t - (W_{t+1} - W_t)\right\| = O\left(\left\|\widetilde{W}_t - W_t\right\| + \hat{\epsilon}'\right).$$

Therefore,

$$\left\|\widetilde{W}_t - W_t\right\| \le C_1^t \hat{\epsilon}',$$

where $C_1$ is a constant larger than 1. So for any $t \le C\log(1/\hat{\epsilon})$, we have

$$\left\|\widetilde{W}_t - W_t\right\| \le C_1^{C\log(1/\hat{\epsilon})}\hat{\epsilon}' \le (1/\hat{\epsilon})^{C_2}\hat{\epsilon}',$$

for some positive constant $C_2$. Choosing $\hat{\epsilon}' = \hat{\epsilon}^{C_2+1}$, we know as long as $n \ge \text{poly}(d, 1/\hat{\epsilon})$, with probability at least 0.99, for any $t \le C\log(1/\hat{\epsilon})$, we have

$$\left\|\widetilde{W}_t - W_t\right\| \le \hat{\epsilon}.$$

$\square$

### B.3 SAMPLE COMPLEXITY ON DOWN-STREAM TASKS

In this section, we give a proof for Theorem 3, which shows that the learned representations can indeed reduce sample complexity in downstream tasks.

**Theorem 3.** *Suppose the downstream data distribution is as defined in Assumption 3. Suppose $\left\|\hat{P} - P\right\|_F \le \hat{\epsilon}$ with $\hat{\epsilon} < 1$. Choose the regularizer coefficient $\rho = \hat{\epsilon}^{1/3}$. For any $\zeta < 1/2$, given $n \ge O(r + \log(1/\zeta))$ number of samples, with probability at least $1 - \zeta$, the training loss minimizer $\hat{w}$ satisfies*

$$\left\|\hat{P}\hat{w} - w^*\right\| \le O\left(\hat{\epsilon}^{1/3} + \beta\frac{\sqrt{r} + \sqrt{\log(1/\zeta)}}{\sqrt{n}}\right).$$

Suppose $\{(x^{(i)}, y^{(i)})\}_{i=1}^n$ are $n$ training samples in the downstream task, let $X \in \mathbb{R}^{n \times d}$ be the data matrix with its $i$-th row equal to $x^{(i)}$. Denote $y \in \mathbb{R}^n$ as the label vector with its $i$-th entry as $y^{(i)}$. Each input $x^{(i)}$ is transformed by a matrix $\hat{P} \in \mathbb{R}^{d \times d}$ to get its representation $\hat{P}x^{(i)}$. The regularized loss can be written as

$$L(w) := \frac{1}{2n}\left\|X\hat{P}w - y\right\|^2 + \frac{\rho}{2}\|w\|^2.$$

This is the ridge regression problem on inputs $\{(\hat{P}x^{(i)}, y^{(i)})\}_{i=1}^n$, and the unique global minimizer $\hat{w}$ has the following close form:

$$\hat{w} = \left(\frac{1}{n}\hat{P}^\top X^\top X\hat{P} + \rho I\right)^{-1}\frac{1}{n}\hat{P}^\top X^\top y \tag{4}$$

With the above closed form of $\hat{w}$, the proof of Theorem 3 follows by bounding the difference between $\hat{P}\hat{w}$ and $w^*$ by matrix concentration inequalities and matrix perturbation bounds. Some proofs of technical lemmas are left in Appendix B.5.

**Proof of Theorem 3.** Denoting $\hat{P}$ as $P + \Delta$, we know $\|\Delta\|_F \le \hat{\epsilon}$ by assumption. We can also write $y$ as $Xw^* + \xi$ where $\xi \in \mathbb{R}^n$ is the noise vector with its $i$-th entry equal to $\xi^{(i)}$. Then, we can divide $\hat{w}$ into two terms,

$$\hat{w} = \left(\frac{1}{n}\hat{P}^\top X^\top X\hat{P} + \rho I\right)^{-1}\frac{1}{n}\hat{P}^\top X^\top y$$

$$= \left(\frac{1}{n}\hat{P}^\top X^\top X\hat{P} + \rho I\right)^{-1}\frac{1}{n}P^\top X^\top (Xw^* + \xi) + \left(\frac{1}{n}\hat{P}^\top X^\top X\hat{P} + \rho I\right)^{-1}\frac{1}{n}\Delta^\top X^\top (Xw^* + \xi)$$

Let's first give an upper bound for the second term that comes from the error term $\Delta^\top$.

**Upper bounding** $\left\| \left( \frac{1}{n} \hat{P}^\top X^\top X \hat{P} + \rho I \right)^{-1} \frac{1}{n} \Delta^\top X^\top \left( X w^* + \xi \right) \right\|$ We first bound the norm of $\frac{1}{n} \Delta^\top X^\top X w^*$. According to Lemma 5, we know with probability at least $1 - \exp(-\Omega(n))$, $\left\| \frac{1}{\sqrt{n}} \Delta^\top X^\top \right\|_F \leq O(\hat{\epsilon})$. Since $X w^*$ is a standard Gaussian vector with dimension $n$, according to Lemma 8, with probability at least $1 - \exp(-\Omega(n))$, $\left\| \frac{1}{\sqrt{n}} X w^* \right\| \leq O(1)$. Therefore, we have $\left\| \frac{1}{n} \Delta^\top X^\top X w^* \right\| \leq O(\hat{\epsilon})$.

Then we bound the norm of $\frac{1}{n} \Delta^\top X^\top \xi$. According to Lemma 8, we know with probability at least $1 - \exp(-\Omega(n))$, $\left\| \frac{1}{\sqrt{n}} \xi \right\| \leq O(\beta)$. According to Lemma 6, we know with probability at least $1 - \zeta/3$, $\left\| \Delta^\top X^\top \bar{\xi} \right\| \leq O\left( \hat{\epsilon} \sqrt{\log(1/\zeta)} \right)$. Therefore, we have $\left\| \frac{1}{n} \Delta^\top X^\top \xi \right\| \leq O\left( \frac{\beta \hat{\epsilon} \sqrt{\log(1/\zeta)}}{\sqrt{n}} \right)$.

Since $\lambda_{\min} \left( \frac{1}{n} \hat{P}^\top X^\top X \hat{P} + \rho I \right) \geq \rho$, we have $\left\| \left( \frac{1}{n} \hat{P}^\top X^\top X \hat{P} + \rho I \right)^{-1} \right\| \leq \frac{1}{\rho}$. Combining with above bound on $\left\| \frac{1}{n} \Delta^\top X^\top \left( X w^* + \xi \right) \right\|$, we know with probability at least $1 - \exp(-\Omega(n)) - \zeta/3$,

$$\left\| \left( \frac{1}{n} \hat{P}^\top X^\top X \hat{P} + \rho I \right)^{-1} \frac{1}{n} \Delta^\top X^\top \left( X w^* + \xi \right) \right\| \leq O\left( \frac{\hat{\epsilon}}{\rho} + \frac{\beta \hat{\epsilon} \sqrt{\log(1/\zeta)}}{\rho \sqrt{n}} \right).$$

**Analyzing** $\left( \frac{1}{n} \hat{P}^\top X^\top X \hat{P} + \rho I \right)^{-1} \frac{1}{n} P^\top X^\top \left( X w^* + \xi \right)$ We can write $\frac{1}{n} \hat{P}^\top X^\top X \hat{P}$ as $\frac{1}{n} P^\top X^\top X P + E$, where

$$E = \frac{1}{n} \Delta^\top X^\top X P + \frac{1}{n} P^\top X^\top X \Delta + \frac{1}{n} \Delta^\top X^\top X \Delta.$$

Let's first bound the spectral norm of $XP$. Since $P$ is a projection matrix on an $r$-dimensional subspace $S$, we can write $P$ as $UU^\top$, where $U \in \mathbb{R}^{d \times r}$ has columns as an orthonormal basis of subspace $S$. According to Lemma 4, we know with probability at least $1 - \exp(-\Omega(n))$,

$$\Omega(1) \leq \sigma_{\min} \left( \frac{1}{\sqrt{n}} X U \right) \leq \sigma_{\max} \left( \frac{1}{\sqrt{n}} X U \right) \leq O(1).$$

Since $\|U\| \leq 1$, we have $\left\| \frac{1}{\sqrt{n}} X P \right\| = \left\| \frac{1}{\sqrt{n}} X U U^\top \right\| \leq O(1)$.

According to Lemma 5, we know with probability at least $1 - \exp(-\Omega(n))$,

$$\left\| \frac{1}{\sqrt{n}} X \Delta \right\|_F \leq O(\hat{\epsilon}).$$

So overall, we know $\|E\| \leq \|E\|_F \leq O(\hat{\epsilon})$.

Then, we can write

$$\left( \frac{1}{n} \hat{P}^\top X^\top X \hat{P} + \rho I \right)^{-1} = \left( \frac{1}{n} P^\top X^\top X P + \rho I \right)^{-1} + F.$$

According to the perturbation bound for matrix inverse (Lemma 11), we have $\|F\| \leq O(\frac{\hat{\epsilon}}{\rho^2})$. Then, we have

$$\left( \frac{1}{n} \hat{P}^\top X^\top X \hat{P} + \rho I \right)^{-1} \frac{1}{n} P^\top X^\top \left( X w^* + \xi \right) = \left( \frac{1}{n} P^\top X^\top X P + \rho I \right)^{-1} \frac{1}{n} P^\top X^\top X w^*$$
$$+ F \frac{1}{n} P^\top X^\top X w^*$$
$$+ \left( \left( \frac{1}{n} P^\top X^\top X P + \rho I \right)^{-1} + F \right) \frac{1}{n} P^\top X^\top \xi$$

We first show that the first term is close to $w^*$. Let the eigenvalue decomposition of $\frac{1}{n}P^\top X^\top XP$ be $V\Sigma V^\top$, where $V$'s columns are an orthonormal basis for subspace $S$. Here $\Sigma \in \mathbb{R}^{r\times r}$ is the diagonal matrix that contains all the eigenvalues of $\frac{1}{n}P^\top X^\top XP$. According to Lemma 4, we know that with probability at least $1 - \exp(-\Omega(n))$, all the non-zero eigenvalues of $\frac{1}{n}P^\top X^\top XP$ are $\Theta(1)$.

Then, it's not hard to show that

$$\left\|\left(\frac{1}{n}P^\top X^\top XP + \rho I\right)^{-1}\frac{1}{n}P^\top X^\top XP - P\right\| \le O(\rho).$$

This immediately implies that

$$\left\|\left(\frac{1}{n}P^\top X^\top XP + \rho I\right)^{-1}\frac{1}{n}P^\top X^\top Xw^* - w^*\right\| \le O(\rho)$$

Next, we bound the norm of the second term $F\frac{1}{n}P^\top X^\top Xw^*$. Similar as before, we know with probability at least $1 - \exp(-\Omega(n))$, $\left\|\frac{1}{\sqrt{n}}Xw^*\right\| \le O(1)$ and $\left\|\frac{1}{\sqrt{n}}P^\top X^\top\right\| \le O(1)$. Therefore, we have

$$\left\|F\frac{1}{n}P^\top X^\top Xw^*\right\| \le \|F\|\left\|\frac{1}{\sqrt{n}}P^\top X^\top\right\|\left\|\frac{1}{\sqrt{n}}Xw^*\right\| \le O\left(\frac{\hat{\epsilon}}{\rho^2}\right).$$

Finally, let's bound the third term $\left(\left(\frac{1}{n}P^\top X^\top XP + \rho I\right)^{-1} + F\right)\frac{1}{n}P^\top X^\top\xi$. We first bound the norm of $\frac{1}{n}P^\top X^\top\xi$. with probability at least $1 - \exp(-\Omega(n))$, we know $\|\xi\| \le 2\beta\sqrt{n}$. Therefore, we know $\left\|\frac{1}{n}P^\top X^\top\xi\right\| \le O(\beta/\sqrt{n})\left\|P^\top X^\top\bar{\xi}\right\|$, where $\bar{\xi} = \xi/\|\xi\|$. According to Lemma 7, with probability at least $1 - \zeta/3$, we have $\left\|P^\top X^\top\bar{\xi}\right\| \le \sqrt{r} + O(\sqrt{\log(1/\zeta)})$. Overall, with probability at least $1 - \exp(-\Omega(n)) - \zeta/3$,

$$\left\|\frac{1}{n}P^\top X^\top\xi\right\| \le O\left(\frac{\sqrt{r}\beta + \sqrt{\log(1/\zeta)}\beta}{\sqrt{n}}\right).$$

It's not hard to verify that for any vector $v \in \mathbb{R}^d$ in the subspace $S$, we have $\left\|\left(\left(\frac{1}{n}P^\top X^\top XP + \rho I\right)^{-1} + F\right)v\right\| \le O(\|v\|)$. Since $\frac{1}{n}P^\top X^\top\xi$ lies on subspace $S$, we have

$$\left\|\left(\left(\frac{1}{n}P^\top X^\top XP + \rho I\right)^{-1} + F\right)\frac{1}{n}P^\top X^\top\xi\right\| \le O\left(\frac{\sqrt{r}\beta + \sqrt{\log(1/\zeta)}\beta}{\sqrt{n}}\right).$$

Combining the above analysis and taking a union bound over all the events, we know with probability at least $1 - \exp(-\Omega(n)) - 2\zeta/3$,

$$\|\hat{w} - w^*\| = O\left(\rho + \frac{\hat{\epsilon}}{\rho} + \frac{\hat{\epsilon}}{\rho^2} + \frac{\beta\hat{\epsilon}\sqrt{\log(1/\zeta)}}{\rho\sqrt{n}} + \frac{\sqrt{r}\beta + \sqrt{\log(1/\zeta)}\beta}{\sqrt{n}}\right)$$

Suppose $n \ge O(\log(1/\zeta))$ and setting $\rho = \hat{\epsilon}^{1/3}$, we further have with probability at least $1 - \zeta$,

$$\|\hat{w} - w^*\| = O\left(\hat{\epsilon}^{1/3} + \frac{\beta\hat{\epsilon}^{2/3}\sqrt{\log(1/\zeta)}}{\sqrt{n}} + \frac{\sqrt{r}\beta + \sqrt{\log(1/\zeta)}\beta}{\sqrt{n}}\right)$$

$$\le O\left(\hat{\epsilon}^{1/3} + \beta\frac{\sqrt{r} + \sqrt{\log(1/\zeta)}}{\sqrt{n}}\right),$$

where the last inequality assumes $\hat{\epsilon} < 1$.

We can also bound $\left\| \hat{P}\hat{w} - w^* \right\|$ as follows,

$$
\begin{aligned}
\left\| \hat{P}\hat{w} - w^* \right\| &= \left\| \hat{P}\hat{w} - P\hat{w} + P\hat{w} - Pw^* \right\| \\
&\leq \left\| \hat{P}\hat{w} - P\hat{w} \right\| + \left\| P\hat{w} - Pw^* \right\| \\
&\leq \left\| \hat{P} - P \right\| \|\hat{w}\| + \|P\| \|\hat{w} - w^*\| \\
&\leq \hat{\epsilon} O\left( 1 + \hat{\epsilon}^{1/3} + \beta \frac{\sqrt{r} + \sqrt{\log(1/\zeta)}}{\sqrt{n}} \right) + O\left( \hat{\epsilon}^{1/3} + \beta \frac{\sqrt{r} + \sqrt{\log(1/\zeta)}}{\sqrt{n}} \right) \\
&\leq O\left( \hat{\epsilon}^{1/3} + \beta \frac{\sqrt{r} + \sqrt{\log(1/\zeta)}}{\sqrt{n}} \right)
\end{aligned}
$$

$\square$

### B.4 ANALYSIS WITH $W_p := (W\mathbb{E}_{x_1} x_1 x_1^\top W^\top)^\alpha$

In this section, we prove that DirectSet($\alpha$) can also learn the projection matrix when we set $W_p := (W\mathbb{E}_{x_1} x_1 x_1^\top W^\top)^\alpha$. For the network architecture and data distribution, we follow exactly the same setting as in Section 3.2. Therefore, we know $W_p := (W\mathbb{E}_{x_1} x_1 x_1^\top W^\top)^\alpha = (W(I + \sigma^2 P_B)W^\top)^\alpha$.

**Theorem 4.** *Suppose network architecture and data distribution are as defined in Assumption 1 and Assumption 2, respectively. Suppose we initialize online network $W$ as $\delta I$, and run DirectPred($\alpha$) on population loss (see Eqn. 1) with infinitesimal step size and $\eta$ weight decay. Suppose we set $W_a = W$ and $W_p = (W\mathbb{E}_{x_1} x_1 x_1^\top W^\top)^\alpha$. Assuming the weight decay coefficient $\eta \in \left( \frac{1}{4(1+\sigma^2)^{1+2\alpha}}, \frac{1}{4} \right)$ and initialization scale $\delta > \left( \frac{1-\sqrt{1-4\eta}}{2} \right)^{1/(2\alpha)}$, we know $W$ converges to $\left( \frac{1+\sqrt{1-4\eta}}{2} \right)^{1/(2\alpha)} P_S$ when time goes to infinity.*

The only difference from Theorem 4 is that now the initialization $\delta$ is only required to be larger than $\frac{1}{4(1+\sigma^2)^{1+2\alpha}}$. The proof is almost the same as in Theorem 1.

**Proof of Theorem 4.** Similar as in the proof of Theorem 1, we can write the dynamics on $W$ is as follows:

$$
\begin{aligned}
\dot{W} &= W_p^\top (-W_p W(I + \sigma^2 P_B) + W_a) - \eta W \\
&= \left| W^2(I + \sigma^2 P_B) \right|^\alpha \left( -\left| W^2(I + \sigma^2 P_B) \right|^\alpha W(I + \sigma^2 P_B) + W \right) - \eta W \\
&= W\left( -(I + \sigma^2 P_B)^{1+2\alpha} |W|^{4\alpha} + |W|^{2\alpha} - \eta \right).
\end{aligned}
$$

**Dynamics for $\lambda_B$:** We can write down the dynamics for $\lambda_B$ as follows:

$$
\dot{\lambda}_B = \lambda_B \left[ -(1+\sigma^2)^{1+2\alpha} |\lambda_B|^{4\alpha} + |\lambda_B|^{2\alpha} - \eta \right]
$$

When $\eta > \frac{1}{4(1+\sigma^2)^{1+2\alpha}}$, we know $\dot{\lambda}_B < 0$ for any $\lambda_B > 0$ and $\lambda_B = 0$ is a critical point. This means, as long as $\eta > \frac{1}{4(1+\sigma^2)^{1+2\alpha}}$, $\lambda_B$ must converge to zero.

**Dynamics for $\lambda_S$:** The dynamics is same as when setting $W_p = (WW^\top)^\alpha$,

$$
\dot{\lambda}_S = \lambda_S \left[ -|\lambda_S|^{4\alpha} + |\lambda_S|^{2\alpha} - \eta \right].
$$

so when $0 < \eta < \frac{1}{4}$ and initialization $\delta^{2\alpha} > \frac{1-\sqrt{1-4\eta}}{2}$, we know $\lambda_S^{2\alpha}$ converges to $\frac{1+\sqrt{1-4\eta}}{2}$.

Overall, we know when $\frac{1}{4(1+\sigma^2)^{1+2\alpha}} < \eta < \frac{1}{4}$ and $\delta > \left( \frac{1-\sqrt{1-4\eta}}{2} \right)^{1/(2\alpha)}$, we have $\lambda_B$ converge to zero and $\lambda_S$ converge to $\left( \frac{1+\sqrt{1-4\eta}}{2} \right)^{1/(2\alpha)}$. That is, matrix $W$ converges to $\left( \frac{1+\sqrt{1-4\eta}}{2} \right)^{1/(2\alpha)} P_S$.

$\square$

B.5 TECHNICAL LEMMAS

**Lemma 3.** *Suppose $\{x^{(i)}, x_1^{(i)}, x_2^{(i)}\}_{i=1}^n$ are sampled as decribed in Section 3. Suppose $n \geq O(d/\hat{\epsilon}^2)$, with probability at least $1 - O(d^2) \exp\left(-\Omega(\hat{\epsilon}^2 n/d^2)\right)$, we have*

$$\left\| \frac{1}{n} \sum_{i=1}^n x_1^{(i)} [x_1^{(i)}]^\top - I - \sigma^2 P_B \right\|, \left\| \frac{1}{n} \sum_{i=1}^n x_1^{(i)} [x_2^{(i)}]^\top - I \right\|, \left\| \frac{1}{n} \sum_{i=1}^n x^{(i)} [x^{(i)}]^\top - I \right\| \leq \hat{\epsilon}.$$

**Proof of Lemma 3.** For each $x_1^{(i)}$, we can write it as $x^{(i)} + z_1^{(i)}$ where $x^{(i)} \sim \mathcal{N}(0, I)$ and $z_1^{(i)} \sim \mathcal{N}(0, \sigma^2 P_B)$. So we have

$$\frac{1}{n} \sum_{i=1}^n x_1^{(i)} [x_1^{(i)}]^\top = \frac{1}{n} \sum_{i=1}^n \left( x^{(i)} [x^{(i)}]^\top + z_1^{(i)} [z_1^{(i)}]^\top + x^{(i)} [z_1^{(i)}]^\top + z_1^{(i)} [x^{(i)}]^\top \right).$$

According to Lemma 9, we know as long as $n \geq O(d/\hat{\epsilon}^2)$, with probability at least $1 - \exp(-\Omega(\hat{\epsilon}^2 n))$,

$$\left\| \frac{1}{n} \sum_{i=1}^n x^{(i)} [x^{(i)}]^\top - I \right\| \leq \hat{\epsilon}.$$

Similarly, with probability at least $1 - \exp(-\Omega(\hat{\epsilon}^2 n))$,

$$\left\| \frac{1}{n} \sum_{i=1}^n z_1^{(i)} [z_1^{(i)}]^\top - \sigma^2 P_B \right\| \leq \hat{\epsilon}.$$

Next we bound $\left\| \frac{1}{n} \sum_{i=1}^n x^{(i)} [z_1^{(i)}]^\top \right\|$. We know each entry in matrix $\frac{1}{n} \sum_{i=1}^n x^{(i)} [z_1^{(i)}]^\top$ is the average of $n$ zero-mean $O(1)$-subexponential independent random variables. Therefore, according to the Bernstein's inequality, for any fixed entry $(k, l)$, with probability at least $1 - \exp\left(-\hat{\epsilon}^2 n/d^2\right)$,

$$\left| \left[ \frac{1}{n} \sum_{i=1}^n x^{(i)} [z_1^{(i)}]^\top \right]_{k,l} \right| \leq \hat{\epsilon}/d.$$

Taking a union bound over all the entries, we know with probability at least $1 - d^2 \exp\left(-\hat{\epsilon}^2 n/d^2\right)$,

$$\left\| \frac{1}{n} \sum_{i=1}^n x^{(i)} [z_1^{(i)}]^\top \right\| \leq \left\| \frac{1}{n} \sum_{i=1}^n x^{(i)} [z_1^{(i)}]^\top \right\|_F \leq \hat{\epsilon}.$$

The same analysis also applies to $\left\| \frac{1}{n} \sum_{i=1}^n z_1^{(i)} [x^{(i)}]^\top \right\|$. Combing all the bounds, we know with probability at least $1 - O(d^2) \exp\left(-\Omega(\hat{\epsilon}^2 n/d^2)\right)$,

$$\left\| \frac{1}{n} \sum_{i=1}^n x_1^{(i)} [x_1^{(i)}]^\top - I - \sigma^2 P_B \right\| \leq 4\hat{\epsilon}.$$

Similarly, we can prove that with probability at least $1 - O(d^2) \exp\left(-\Omega(\hat{\epsilon}^2 n/d^2)\right)$,

$$\left\| \frac{1}{n} \sum_{i=1}^n x_1^{(i)} [x_2^{(i)}]^\top - I \right\| \leq 4\hat{\epsilon}.$$

Changing $\hat{\epsilon}$ to $\hat{\epsilon}'/4$ finishes the proof. $\qquad\square$

**Lemma 4.** *Let $X \in \mathbb{R}^{n \times d}$ be a standard Gaussian matrix, and let $U \in \mathbb{R}^{d \times r}$ be a matrix with orthonormal columns. Suppose $n \geq 2r$, with probability at least $1 - \exp(-\Omega(n))$, we know*

$$\Omega(1) \leq \lambda_{\min}\left( \frac{1}{n} U^\top X^\top X U \right) \leq \lambda_{\max}\left( \frac{1}{n} U^\top X^\top X U \right) \leq O(1).$$

**Proof of Lemma 4.** Since $U$ has orthonormal columns, we know $XU$ is a $n \times r$ matrix with each entry independently sampled from $\mathcal{N}(0, 1)$. According to Lemma 9, we know when $n \geq 2r$, with probability at least $1 - \exp(-\Omega(n))$,

$$\Omega(1) \leq \sigma_{\min}\left(\frac{1}{\sqrt{n}}XU\right) \leq \sigma_{\max}\left(\frac{1}{\sqrt{n}}XU\right) \leq O(1).$$

This immediately implies that

$$\Omega(1) \leq \lambda_{\min}\left(\frac{1}{n}U^\top X^\top XU\right) \leq \lambda_{\max}\left(\frac{1}{n}U^\top X^\top XU\right) \leq O(1).$$

$\square$

**Lemma 5.** *Let $\Delta$ be a $d \times d$ matrix with Frobenius norm $\hat{\epsilon}$, and let $X$ be a $n \times d$ standard Gaussian matrix. We know with probability at least $1 - \exp(-\Omega(n))$,*

$$\left\|\frac{1}{\sqrt{n}}X\Delta\right\|_F \leq O(\hat{\epsilon}).$$

**Proof of Lemma 5.** Let the singular value decomposition of $\Delta$ be $U\Sigma V^\top$, where $U, V$ have orthonormal columns and $\Sigma$ is a diagonal matrix with diagonals equal to singular values $\sigma_i$'s. Since $\|\Delta\|_F = \hat{\epsilon}$, we know $\sum_{i=1}^{d} \sigma_i^2 = \hat{\epsilon}^2$.

Since $U$ is an orthonormal matrix, we know $\hat{X} := XU$ is still an $n \times d$ standard Gaussian matrix. Next, we bound the Frobenius norm of $\widetilde{X} := \hat{X}\Sigma$. It's not hard to verify that all the entries in $\widetilde{X}$ are independent Gaussian variables and $\widetilde{X}_{ij} \sim \mathcal{N}(0, \sigma_j^2)$. According to the Bernstein's inequality for sum of independent and sub-exponential random variables, we have for every $t > 0$,

$$\Pr\left[\left|\sum_{i \in [n], j \in [d]} \widetilde{X}_{ij}^2 - n\hat{\epsilon}^2\right| \geq t\right] \leq 2\exp\left[-c\min\left(\frac{t^2}{\sum_{i \in [n], j \in [d]} \sigma_j^4}, \frac{t}{\max_{j \in [d]} \sigma_j^2}\right)\right].$$

Since $\sum_{j=1}^{d} \sigma_j^2 = \|\Delta\|_F^2 = \hat{\epsilon}^2$, we know $\max_{j \in [d]} \sigma_j^2 \leq \hat{\epsilon}^2$. We also have $\sum_{j \in [d]} \sigma_j^4 \leq \left(\sum_{j \in [d]} \sigma_j^2\right)^2 = \hat{\epsilon}^4$. Therefore, we have

$$\Pr\left[\left|\sum_{i \in [n], j \in [d]} \widetilde{X}_{ij}^2 - n\hat{\epsilon}^2\right| \geq t\right] \leq 2\exp\left[-c\min\left(\frac{t^2}{n\hat{\epsilon}^4}, \frac{t}{\hat{\epsilon}^2}\right)\right].$$

Replacing $t$ by $n\hat{\epsilon}^2$, we concluded that with probability at least $1 - \exp(-\Omega(n))$, $\left\|\widetilde{X}\right\|_F^2 \leq 2n\hat{\epsilon}^2$. Furthermore, since $\|V^\top\| = 1$, we have

$$\left\|\frac{1}{\sqrt{n}}X\Delta\right\|_F = \left\|\frac{1}{\sqrt{n}}\widetilde{X}V^\top\right\|_F \leq \left\|\frac{1}{\sqrt{n}}\widetilde{X}\right\|_F \|V\| \leq O(\hat{\epsilon}).$$

$\square$

**Lemma 6.** *Let $\Delta^\top$ be a $d \times d$ matrix with Frebenius norm $\hat{\epsilon}$ and let $X^\top$ be a $d \times n$ standard Gaussian matrix. Let $\bar{\xi}$ be a unit vector with dimension $n$. We know with probability at least $1 - \zeta/3$,*

$$\left\|\Delta^\top X^\top \bar{\xi}\right\| \leq O(\hat{\epsilon}\sqrt{\log(1/\zeta)}).$$

**Proof of Lemma 6.** Let the sigular value decomposition of $\Delta^\top$ be $U\Sigma V^\top$. We know $X^\top \bar{\xi}$ is a $d$-dimensional standard Gaussian vector. Further, we know $V^\top X^\top \bar{\xi}$ is also a $d$-dimensional standard Gaussian vector. So $\Sigma V^\top X^\top \bar{\xi}$ has independent Gaussian entries with its $i$-th entry distributed as $\mathcal{N}(0, \sigma_i^2)$. According to the Bernstein's inequality for sum of independent and sub-exponential random variables, we have for every $t > 0$,

$$\Pr\left[\left|\|\Sigma V^\top X^\top \bar{\xi}\|^2 - \hat{\epsilon}^2\right| \geq t\right] \leq 2\exp\left[-c\min\left(\frac{t^2}{\hat{\epsilon}^4}, \frac{t}{\hat{\epsilon}^2}\right)\right].$$

Choosing $t$ as $O(\hat{\epsilon}^2 \log(1/\zeta))$, we know with probability at least $1 - \zeta/3$, we have

$$\left\| \Sigma V^\top X^\top \bar{\xi} \right\|^2 \leq O\left(\hat{\epsilon}^2 \log(1/\zeta)\right).$$

Since $\|U\| = 1$, we further have

$$\left\| \Delta^\top X^\top \bar{\xi} \right\| = \left\| U \Sigma V^\top X^\top \bar{\xi} \right\| \leq \|U\| \left\| \Sigma V^\top X^\top \bar{\xi} \right\| \leq O\left(\hat{\epsilon} \sqrt{\log(1/\zeta)}\right)$$

$\square$

**Lemma 7.** *Let $P \in \mathbb{R}^{d \times d}$ be a projection matrix on a $r$-dimensional subspace, and let $\bar{\xi}$ be a unit vector in $\mathbb{R}^d$. Let $X^\top$ be a $d \times n$ standard Gaussian matrix that is independent with $P$ and $\xi$. With probability at least $1 - \zeta/3$, we have*

$$\left\| P^\top X^\top \bar{\xi} \right\| \leq \sqrt{r} + O(\sqrt{\log(1/\zeta)}).$$

**Proof of Lemma 7.** Since $P$ is a projection matrix on an $r$-dimensional subspace, we can write $P$ as $UU^\top$, where $U \in \mathbb{R}^{d \times r}$ has orthonormal columns. We know $U^\top X^\top$ is still a standard Gaussian matrix with dimension $r \times n$. Furthermore, $U^\top X^\top \bar{\xi}$ is an $r$-dimensional standard Gaussian vector. According to Lemma 8, with probability at least $1 - \zeta/3$, we have

$$\left\| U^\top X^\top \bar{\xi} \right\| \leq \sqrt{r} + O(\sqrt{\log(1/\zeta)}).$$

Since $\|U\| = 1$, we further have

$$\left\| P^\top X^\top \bar{\xi} \right\| = \left\| UU^\top X^\top \bar{\xi} \right\| \leq \|U\| \left\| U^\top X^\top \bar{\xi} \right\| \leq \sqrt{r} + O(\sqrt{\log(1/\zeta)}).$$

$\square$

## C  ANALYSIS OF DEEP LINEAR NETWORKS

In this section, we extend the analysis in Section 3.2 to deep linear networks. We consider the same data distribution as defined in Assumption 2. We consider the following network,

**Assumption 4** (Deep linear network)**.** *The online network is an $l$-layer linear networks $W_l W_{l-1} \cdots W_1$ with each $W_i \in \mathbb{R}^{d \times d}$. The target network has the same architecture with weight matrices $W_{a,l} W_{a,l-1} \cdots W_{a,1}$. For convenience, we denote $W$ as $W_l W_{l-1} \cdots W_1$ and denote $W_a$ as $W_{a,l} W_{a,l-1} \cdots W_{a,1}$.*

**Training procedure:**  At the initialization, we initialize each $W_i$ as $\delta^{1/l} I_d$. Through the training, we fix $W_p$ as $\left(WW^\top\right)^\alpha$ and fix each $W_{a,i}$ as $W_i$. We run gradient flow on every $W_i$ with weight decay $\eta$. The population loss is

$$L(\{W_i\}, W_p, \{W_{a,i}\}) := \frac{1}{2} \mathbb{E}_{x_1, x_2} \left\| W_p W_l W_{l-1} \cdots W_1 x_1 - \text{StopGrad}(W_{a,l} W_{a,l-1} \cdots W_{a,1} x_2) \right\|^2.$$

**Theorem 5.** *Suppose the data distribution and network architecture satisfies Assumption 2 and Assumption 4, respectively. Suppose we train the network as described above. Assuming the weight decay coefficient*

$$\eta \in \left( \frac{2\alpha l(2\alpha l + 2l - 2)^{1 + \frac{1}{\alpha} - \frac{1}{\alpha l}}}{(4\alpha l + 2l - 2)^{2 + \frac{1}{\alpha} - \frac{1}{\alpha l}}(1 + \sigma^2)^{1 + \frac{1}{\alpha} - \frac{1}{\alpha l}}}, \frac{2\alpha l(2\alpha l + 2l - 2)^{1 + \frac{1}{\alpha} - \frac{1}{\alpha l}}}{(4\alpha l + 2l - 2)^{2 + \frac{1}{\alpha} - \frac{1}{\alpha l}}} \right), \text{ and initialization scale } \delta \geq$$

$\left( \frac{2\alpha l + 2l - 2}{4\alpha l + 2l - 2} \right)^{\frac{1}{2\alpha}}$, *we know $W$ converges to $cP_S$ as time goes to infinity, where $c$ is a positive number within $\left( \left( \frac{2\alpha l + 2l - 2}{4\alpha l + 2l - 2} \right)^{\frac{1}{2\alpha}}, 1 \right)$.*

Similar as in the setting of single-layer linear networks, we prove Theorem 5 by analyzing the dynamics of the eigenvalues of $W$. Note that with constant $\alpha$, the upper/lower bounds for $\eta$ and scalar $c$ in the Theorem are always constants no matter how large $l$ is.

**Proof of Theorem 5.** For $j \geq i$, we use $W_{[j:i]}$ to denote $W_j W_{j-1} \cdots W_i$ and for $j < i$ have $W_{[j:i]} = I$. We use similar notations for $W_{a,[j:i]}$. For each $W_i$, we can compute its dynamics as follows:

$$\dot{W}_i = -\left(W_p W_{[l:i+1]}\right)^\top \left(W_p W(I + \sigma^2 P_B)\right)\left(W_{[i-1:1]}\right)^\top + \left(W_p W_{a,[l:i+1]}\right)^\top W_a \left(W_{a,[i-1:1]}\right)^\top - \eta W_i.$$

It's clear that through the training all $W_i$'s remains the same and they are simultaneously diagonalizable with $W_p$, $I$ and $P_B$. We also have $W_a = W$ and $W_p = |W|^{2\alpha}$. Since we will ensure that $W$ is always positive semi-definite so $W_p = |W|^{2\alpha} = W^{2\alpha} = W_i^{2\alpha l}$. So the dynamics for each $W_i$ can be simplified as follows:

$$\dot{W}_i = -W_i^{4\alpha l + 2l - 1}(I + \sigma^2 P_B) + W_i^{2\alpha l + 2l - 1} - \eta W_i.$$

Let the eigenvalue decomposition of $W_i$ be $\sum_{i=1}^d \nu_i u_i u_i^\top$, with $\text{span}(\{u_{d-r+1}, \cdots, u_d\})$ equals to subspace $B$. We can separately analyze the dynamics of each $\nu_i$. Furthermore, we know $\nu_1, \cdots, \nu_r$ have the same value $\nu_S$ and $\nu_{d-r+1}, \cdots, \nu_d$ have the same value $\nu_B$. We can write down the dynamics for $\nu_S$ and $\nu_B$ as follows,

$$\dot{\nu}_S = -\nu_S^{4\alpha l + 2l - 1} + \nu_S^{2\alpha l + 2l - 1} - \eta \nu_S,$$
$$\dot{\nu}_B = -\nu_B^{4\alpha l + 2l - 1}(1 + \sigma^2) + \nu_B^{2\alpha l + 2l - 1} - \eta \nu_B.$$

Let $\lambda_S$ be the eigenvalue of $W$ corresponding to eigen-directions $u_1, \cdots, u_r$, and let $\lambda_B$ be the eigenvalue of $W$ corresponding to eigen-directions $u_{d-r+1}, \cdots, u_d$. We know $\lambda_S = \nu_S^l$ and $\lambda_B = \nu_B^l$. So we can write down the dynamics for $\lambda_B$ as follows,

$$\dot{\lambda}_B = l\nu_B^{l-1}\dot{\nu}_B = -l\nu_B^{4\alpha l + 3l - 2}(1 + \sigma^2) + l\nu_B^{2\alpha l + 3l - 2} - l\eta \nu_B^l$$
$$= -l\lambda_B^{4\alpha + 3 - 2/l}(1 + \sigma^2) + l\lambda_B^{2\alpha + 3 - 2/l} - l\eta \lambda_B,$$

and similarly for $\lambda_S$ we have

$$\dot{\lambda}_S = -l\lambda_S^{4\alpha + 3 - 2/l} + l\lambda_S^{2\alpha + 3 - 2/l} - l\eta \lambda_S.$$

**Dynamics for $\lambda_B$:** We can write the dynamics on $\lambda_B$ as follows,

$$\dot{\lambda}_B = l\lambda_B g(\lambda_B),$$

where $g(\lambda_B) := -\lambda_B^{4\alpha + 2 - 2/l}(1 + \sigma^2) + \lambda_B^{2\alpha + 2 - 2/l} - \eta$. We show that when $\eta$ is large enough, $g(\lambda_B)$ is negative for any positive $\lambda_B$. We compute the maximum value of $g(\lambda_B)$ for $\lambda_B > 0$. We first compute the derivative of $g$ as follows:

$$g'(\lambda_B) = -(4\alpha + 2 - 2/l)(1 + \sigma^2)\lambda_B^{4\alpha + 1 - 2/l} + (2\alpha + 2 - 2/l)\lambda_B^{2\alpha + 1 - 2/l}$$
$$= \lambda_B^{2\alpha + 1 - 2/l}\left(-(4\alpha + 2 - 2/l)(1 + \sigma^2)\lambda_B^{2\alpha} + (2\alpha + 2 - 2/l)\right).$$

It's clear that $g'(\lambda_B) > 0$ for $\lambda_B^{2\alpha} \in (0, \frac{2\alpha l + 2l - 2}{(4\alpha l + 2l - 2)(1 + \sigma^2)})$ and $g'(\lambda_B) < 0$ for $\lambda_B^{2\alpha} \in (\frac{2\alpha l + 2l - 2}{(4\alpha l + 2l - 2)(1 + \sigma^2)}, +\infty)$. Therefore, the maximum value of $g(\lambda_B)$ for positive $\lambda_B$ takes at $\lambda_B^* = \left(\frac{2\alpha l + 2l - 2}{(4\alpha l + 2l - 2)(1 + \sigma^2)}\right)^{\frac{1}{2\alpha}}$ and

$$g(\lambda_B^*) = -\left(\frac{2\alpha l + 2l - 2}{(4\alpha l + 2l - 2)(1 + \sigma^2)}\right)^{2 + \frac{1}{\alpha} - \frac{1}{\alpha l}}(1 + \sigma^2) + \left(\frac{2\alpha l + 2l - 2}{(4\alpha l + 2l - 2)(1 + \sigma^2)}\right)^{1 + \frac{1}{\alpha} - \frac{1}{\alpha l}} - \eta$$
$$= \frac{2\alpha l(2\alpha l + 2l - 2)^{1 + \frac{1}{\alpha} - \frac{1}{\alpha l}}}{(4\alpha l + 2l - 2)^{2 + \frac{1}{\alpha} - \frac{1}{\alpha l}}(1 + \sigma^2)^{1 + \frac{1}{\alpha} - \frac{1}{\alpha l}}} - \eta.$$

As long as $\eta > \frac{2\alpha l(2\alpha l + 2l - 2)^{1 + \frac{1}{\alpha} - \frac{1}{\alpha l}}}{(4\alpha l + 2l - 2)^{2 + \frac{1}{\alpha} - \frac{1}{\alpha l}}(1 + \sigma^2)^{1 + \frac{1}{\alpha} - \frac{1}{\alpha l}}}$, we know $g(\lambda_B) < 0$ for any $\lambda_B > 0$, which further implies that $\dot{\lambda}_B < 0$ for any $\lambda_B > 0$. So $\lambda_B$ converges to zero.

**Dynamics for $\lambda_S$:** We can write down the dynamics on $\lambda_S$ as follows,

$$\dot{\lambda}_S = l\lambda_S h(\lambda_S),$$

where $h(\lambda_S) = -\lambda_S^{4\alpha + 2 - 2/l} + \lambda_S^{2\alpha + 2 - 2/l} - \eta$. We compute the derivative of $h$ as follows:

$$h'(\lambda_S) = \lambda_S^{2\alpha + 1 - 2/l}\left(-(4\alpha + 2 - 2/l)\lambda_S^{2\alpha} + (2\alpha + 2 - 2/l)\right).$$

So $h(\lambda_S)$ is increasing in $(0, \left(\frac{2\alpha l+2l-2}{4\alpha l+2l-2}\right)^{\frac{1}{2\alpha}})$ and is decreasing in $(\left(\frac{2\alpha l+2l-2}{4\alpha l+2l-2}\right)^{\frac{1}{2\alpha}}, \infty)$. The maximum value of $h$ for positive $\lambda_S$ takes at $\lambda_S^* = \left(\frac{2\alpha l+2l-2}{4\alpha l+2l-2}\right)^{\frac{1}{2\alpha}}$ and we have

$$h(\lambda_S^*) = \frac{2\alpha l(2\alpha l + 2l - 2)^{1+\frac{1}{\alpha}-\frac{1}{\alpha l}}}{(4\alpha l + 2l - 2)^{2+\frac{1}{\alpha}-\frac{1}{\alpha l}}} - \eta.$$

As long as $\eta < \frac{2\alpha l(2\alpha l+2l-2)^{1+\frac{1}{\alpha}-\frac{1}{\alpha l}}}{(4\alpha l+2l-2)^{2+\frac{1}{\alpha}-\frac{1}{\alpha l}}}$, we have $h(\lambda_S^*) > 0$. Furthermore, since $h$ is increasing in $(0, \lambda_S^*)$ and is decreasing in $(\lambda_S^*, \infty)$ and $h(0), h(\infty) < 0$, we know there exists $\lambda_S^- \in (0, \lambda_S^*), \lambda_S^+ \in (\lambda_S^*, \infty)$ such that $h(\lambda_S) < 0$ in $(0, \lambda_S^-)$, $h(\lambda_S) > 0$ in $(\lambda_S^-, \lambda_S^+)$ and $h(\lambda_S) < 0$ in $(\lambda_S^+, \infty)$. Therefore, as long as $\delta \geq \lambda_S^* > \lambda_S^-$, we have $\lambda_S$ converges to $\lambda_S^+$. Since $h(1) < 0$, we know $\lambda_S^+ \in (\left(\frac{2\alpha l+2l-2}{4\alpha l+2l-2}\right)^{\frac{1}{2\alpha}}, 1)$.

Overall as long as $\eta \in \left(\frac{2\alpha l(2\alpha l-2)^{1+\frac{1}{\alpha}-\frac{1}{\alpha l}}}{(4\alpha l+2l-2)^{2+\frac{1}{\alpha}-\frac{1}{\alpha l}}(1+\sigma^2)^{1+\frac{1}{\alpha}-\frac{1}{\alpha l}}}, \frac{2\alpha l(2\alpha l+2l-2)^{1+\frac{1}{\alpha}-\frac{1}{\alpha l}}}{(4\alpha l+2l-2)^{2+\frac{1}{\alpha}-\frac{1}{\alpha l}}}\right)$, we know $W$ converges to $cP_S$, where $c$ is a positive number within $(\left(\frac{2\alpha l+2l-2}{4\alpha l+2l-2}\right)^{\frac{1}{2\alpha}}, 1)$. $\qquad\square$

## D ANALYSIS OF PREDICTOR REGULARIZATION.

In this section, we study the influence of predictor regularization in a simple linear setting. In particular, we consider the same setting as in Section 3.2 except that we set $W_p := (WW^\top)^\alpha + \epsilon I$.

**Theorem 6.** *In the setting of Theorem 1 except that we set $W_p = (WW^\top)^\alpha + \epsilon I$. We have*

- *when $\epsilon \in [0, \frac{1+\sqrt{1-4\eta}}{2})$, as long as $\delta > \left(\max\left(\frac{1-\sqrt{1-4\eta}}{2} - \epsilon, 0\right)\right)^{\frac{1}{2\alpha}}$, we have $W$ converges to $\left(\frac{1+\sqrt{1-4\eta}}{2} - \epsilon\right)^{\frac{1}{2\alpha}} P_S$;*

- *when $\epsilon \geq \frac{1+\sqrt{1-4\eta}}{2}$, $W$ always converges to zero.*

**Proof of Theorem 6.** We can write the dynamics of $W$ as follows,

$$\dot{W} = W_p^\top \left(-W_p W(I + \sigma^2 P_B) + W_a\right) - \eta W$$
$$= W \left(-(I + \sigma^2 P_B)\left(|W|^{2\alpha} + \epsilon I\right)^2 + \left(|W|^{2\alpha} + \epsilon I\right) - \eta\right).$$

Let the eigenvalue decomposition of $W$ be $\sum_{i=1}^d \lambda_i u_i u_i^\top$, with $\mathrm{span}(\{u_{d-r+1}, \cdots, u_d\})$ equals to subspace $B$. We can separately analyze the dynamics of each $\lambda_i$. Furthermore, we know $\lambda_1, \cdots, \lambda_r$ have the same value $\lambda_S$ and $\lambda_{d-r+1}, \cdots, \lambda_d$ have the same value $\lambda_B$.

**Dynamics for $\lambda_B$:** We can write down the dynamics for $\lambda_B$ as follows:

$$\dot{\lambda}_B = \lambda_B \left[-(1+\sigma^2)\left(|\lambda_B|^{2\alpha} + \epsilon\right)^2 + \left(|\lambda_B|^{2\alpha} + \epsilon\right) - \eta\right]$$

When $\eta > \frac{1}{4(1+\sigma^2)}$, we still know $\dot{\lambda}_B < 0$ for any $\lambda_B > 0$ and $\lambda_B = 0$ is a critical point. So $\lambda_B$ converges to zero.

**Dynamics for $\lambda_S$:** We can write down the dynamics for $\lambda_S$ as follows:

$$\dot{\lambda}_S = \lambda_S \left[-\left(|\lambda_S|^{2\alpha} + \epsilon\right)^2 + \left(|\lambda_S|^{2\alpha} + \epsilon\right) - \eta\right]$$
$$= -\lambda_S \left(|\lambda_S|^{2\alpha} + \epsilon - \frac{1-\sqrt{1-4\eta}}{2}\right)\left(|\lambda_S|^{2\alpha} + \epsilon - \frac{1+\sqrt{1-4\eta}}{2}\right),$$

where the second inequality assumes $0 < \eta < \frac{1}{4}$. We have

- when $\epsilon \in [0, \frac{1+\sqrt{1-4\eta}}{2})$, as long as $\delta > \left( \max \left( \frac{1-\sqrt{1-4\eta}}{2} - \epsilon, 0 \right) \right)^{\frac{1}{2\alpha}}$, we have $\lambda_S$ converges to $\left( \frac{1+\sqrt{1-4\eta}}{2} - \epsilon \right)^{\frac{1}{2\alpha}} > 0$;

- when $\epsilon \geq \frac{1+\sqrt{1-4\eta}}{2}$, $\lambda_S$ always converges to zero.

$\square$

# E TECHNICAL TOOLS

## E.1 NORM OF RANDOM VECTORS

The following lemma shows that a standard Gaussian vector with dimension $n$ has $\ell_2$ norm concentrated at $\sqrt{n}$.

**Lemma 8** (Theorem 3.1.1 in Vershynin (2018)). *Let $X = (X_1, X_2, \cdots, X_n) \in \mathbb{R}^n$ be a random vector with each entry independently sampled from $\mathcal{N}(0, 1)$. Then*

$$\Pr[|\|x\| - \sqrt{n}| \geq t] \leq 2\exp(-t^2/C^2),$$

*where $C$ is an absolute constant.*

## E.2 SINGULAR VALUES OF GAUSSIAN MATRICES

The following lemma shows a tall random Gaussian matrix is well-conditioned with high probability.

**Lemma 9** (Corollary 5.35 in Vershynin (2010)). *Let $A$ be an $N \times n$ matrix whose entries are independent standard normal random variables. Then for every $t \geq 0$ with probability at least $1 - 2\exp(-t^2/2)$ one has*

$$\sqrt{N} - \sqrt{n} - t \leq s_{\min}(A) \leq s_{\max}(A) \leq \sqrt{N} + \sqrt{n} + t$$

## E.3 PERTURBATION BOUND FOR MATRIX PSEUDO-INVERSE

With a lowerbound on $\sigma_{\min}(A)$, we can get bounds for the perturbation of pseudo-inverse.

**Lemma 10** (Theorem 3.4 in Stewart (1977)). *Consider the perturbation of a matrix $A \in \mathbb{R}^{m \times n}$: $B = A + E$. Assume that $rank(A) = rank(B) = n$, then*

$$\|B^\dagger - A^\dagger\| \leq \sqrt{2} \|A^\dagger\| \|B^\dagger\| \|E\|.$$

The following corollary is particularly useful for us.

**Lemma 11** (Lemma G.8 in Ge et al. (2015)). *Consider the perturbation of a matrix $A \in \mathbb{R}^{m \times n}$: $B = A + E$ where $\|E\| \leq \sigma_{\min}(A)/2$. Assume that $rank(A) = rank(B) = n$, then*

$$\|B^\dagger - A^\dagger\| \leq 2\sqrt{2} \|E\| / \sigma_{\min}(A)^2.$$

