# OpenReview forum: "Towards Demystifying Representation Learning with Non-contrastive Self-supervision"
_ICLR.cc/2022/Conference — ICLR 2022 Submitted_

### Official Review · Reviewer_racS · 2021-10-29

**Correctness:** 4
**Technical Novelty And Significance:** 3
**Empirical Novelty And Significance:** 2
**Recommendation:** 6
**Confidence:** 3

**Main Review:**

- the main paper seems correct. Technically it is strongly inspired by the analysis in Tian et al. 2021. In chapter 2 their algorithm is generalized by the additional exponent of the correlation matrix F. From reading the paper it remains a bit elusive why this is done (apart from the nice consequence that for alpha=1 the method is simpler). It would be great to have add an intuition why this was/is done.
- I see the main contribution of the paper in the analysis of the representation learning on the simple toy data with linear invariant & nuisance spaces, which is nice and novel to my knowledge. This result is not surprising - as all nc-ssl methods are designed to perform exactly this separation. But it's great to have a mathematical analysis on this.
- the empirical analyses follow the same experiments as in Tian et al. 2021, with comparable results. The results of Section 4.2. on ImageNet should be put into a table (and the best methods bolded). Like this it feels like the results are hidden.


Minor Issues.
- I find the point in the abstract "why these methods do not collapse to the trivial solutions" quite strong, as Tiang et al have made quite a good analysis on this. So, I'd suggest to rephrase and focus instead on your contribution (the 2nd point), and refer to this point being already analyzed (instead of having an enumeration).
- Figure 2: please add X axes labels on the leftmost plots. On the rightmost plots: can the eigenvalues corresponding to eigenvectors in the invariant subspace and the ones from the nuisance subspace be visually separated? Right now, one has to assume, instead of know, that the smaller ones are the nuisance ones.
- Theorem 2: eps is overloaded, which might be confusing to the reader. It's already been used as the regularization parameter in the description of the algorithm.
- please check again all verb forms throughout the paper, also singular and plurals are often wrong. Some sentences are very long and could be reduced in length for better readability (a style taste issue of course).

**Summary Of The Paper:**

The paper "Towards demystifying representation learning with non-contrastive self-supervision" presents and analyzes a family of algorithms, DirectSet(\alpha), for non-conctrastive self-supervised learning with only positive pairs. The theoretical analysis assumes linear layers, and focuses on a special data distribution assumption, where the input space is separated in two linear subspaces, one invariant under the 'data augmentations', the other being the complement. It is then shown that the proposed algorithm converges to the projection matrix onto the invariant sub-space. Further it is shown theoretically that this has the down-stream advantage reduced sample complexity for learning on this representation. Empirically, it is shown that the method performs on par, or sometimes slightly better, than the previously proposed closely related method DirectPred (Tian et al. 2021).

**Summary Of The Review:**

The paper seems overall technically correct. It presents a nice analysis that illustrates how the representation learning happens in a toy data setting and proposes a slightly more simple algorithm than in previous work. This is great, though not a major step forward (as ssl methods are designed to do this, the results are expected IMHO). Further, the experimental analyses indicate no significant improvements over previous work.

---

> ### Author Response · Authors · 2021-11-16
> **Response**
>
> Thank you so much for your positive review and valuable suggestions! We are glad to answer your questions as below.
>
> “Why the algorithm is generalized to allow arbitrary exponent on correlation matrix F?”:
>
> Considering DirectSet(alpha) with arbitrary alpha allows us to compare the performance of DirectCopy not only with DirectPred but also with DirectSet(alpha) with other alpha’s. We also proved a more general theory by considering DirectSet(alpha) with general alpha.
>
> “The results of Section 4.2. on ImageNet should be put into a table”:
>
> Thanks for the suggestion! We have included a table for our ImageNet results in the revised version.
>
> We hope our response answers your questions. We also thank you for your valuable suggestions in the writing, we have revised our paper accordingly.

---

### Official Review · Reviewer_G6yG · 2021-11-03

**Correctness:** 4
**Technical Novelty And Significance:** 2
**Empirical Novelty And Significance:** 3
**Recommendation:** 3
**Confidence:** 4

**Main Review:**

*Contributions*:

The authors present a generalized formulation, which they call DirectSet(ꭤ), for setting the weights of the predictor network along the lines of DirectPred (Tian et al) and show that there exists an implicit threshold that governs what features are learnt, based on the weight decay 'η'. They show that for the special case with ꭤ = 1, we can obtain higher top-1 accuracy than BYOL on Imagenet, learn more distinguishable features in polynomial number of samples, and reduce the sample complexity on downstream tasks.

*Strengths*:

The authors present an interesting take on the evolution of invariant and nuisance features throughout training, and how this is acutely affected by the weight decay. Their findings are also well backed-up by experiments: they demonstrate the performance of their method on three different datasets (STL-10, CIFAR-10/100, and Imagenet) along with ablations of important parameters on STL-10 and CIFAR-10. Overall, the paper is well-structured.

*Weaknesses*:
- *Major*: The paper seems to be a derivative of previous work (Tian et al) and hence lacks novelty in my opinion.

*Relevance and impact*:

I think this work presents some interesting analysis with the proposed method DirectSet(ꭤ): how it learns useful features, and reduces sample complexity on downstream tasks, but it is not very clear why it doesn’t collapse to a trivial solution. Moreover, it is unclear how their analysis extends more generally to other non-contrastive SSL methods such as BYOL or Sim-Siam. Based on the above reasoning, in my opinion, the paper doesn’t seem to have a strong impact.

*Comments/Questions*:

1. In the ablation study with varying weight decay, the authors also test out ꭤ = 2, which should learn features that are stronger and more distinguishable (as claimed in section 4). Is this the case? Or do such features not translate to better downstream performance? Why aren’t more cases with ꭤ > 1 explored?
2. Is there some intuition behind why DirectCopy doesn’t perform as well on CIFAR-100, as seen in Table 1?
3. The authors mention in Section 6 that whether the nuisance features come back or not is related to the downstream task performance. Why is this the case?
4. The paper is not well-written and is peppered with multiple typos (including the method being called DirectPred in Theorem 1 instead of DirectSet. Also, the related works section could have been a bit more extensive for better context. Figures can be more informative as well with some labels for the regions demarcated by the basins.



**Summary Of The Paper:**

This paper attempts to investigate the reasons behind why non-contrastive SSL methods such as BYOL and Sim-Siam do not collapse to trivial solutions, how they learn representations that are related to the data distribution and augmentation process, and how these reduce the sample-complexity of downstream tasks. It heavily draws upon DirectPred (Tian et al) and generalizes the method for directly setting the weights of the predictor network via a parameter that seems to be tied to the strength and distinguishability of the learnt features.

Tian et al: https://arxiv.org/abs/2102.06810

**Summary Of The Review:**

I think the paper itself can benefit from clearer writing and some emphasis on how their analysis scales up to explain other non-contrastive SSL methods.

However, if I am to evaluate it in a broader context, I think in its current state it lacks novelty to be seen as a standalone piece of work.

---

> ### Author Response · Authors · 2021-11-16
> **Response**
>
> Thanks a lot for your detailed review and valuable suggestions! We are glad to take this opportunity to address your concerns as below.
>
> “Comparison with Tian et al 2021”:
>
> Tian et al only explained why the representation in nc-SSL does not collapse, but did not study what representation is learned and how the representation is related to the data distribution and augmentation process. Our analysis reveals for the first time that weight decay can serve as a threshold that selectively discards the nuisance features with high variance under augmentation and keeps the invariant features with low variance. We view our paper as a first step towards understanding *how* the representation is learned in nc-SSL.
>
> “It’s not very clear why the representation doesn’t collapse to a trivial solution”:
>
> Theorem 1 shows that the online network converges to a desirable projection matrix, which  *implies* that it does not converge to a trivial zero solution. Basically, the weight decay only discards nuisance features and keeps invariant features, so the representation will not converge to zero.
>
> “How the analysis extends to other non-contrastive SSL methods such as BYOL or Sim-Siam”:
>
> With the predictor updated by gradient methods, BYOL and Sim-Siam are much trickier to analyze. Since DirectPred (or DirectSet) is much more amenable for theoretical analysis and also enjoys comparable or even better performance than BYOL/Sim-Siam, we focus our analysis on DirectPred in this initial work.
>
> It is also very possible that the original BYOL/SimSiam can learn a similar representation as DirectPred (supported by Fig. 2 in Tian et al 2021), but due to the complexity of the gradient descent algorithm, we leave the analysis to future work.
>
> “Cases with alpha>1”:
>
> We tried alpha equals 2 and 4 in our experiments, but it did not lead to better downstream performance compared with alpha=1. This is probably because of the diminishing benefits of stronger and more distinguishable features beyond alpha=1. Using alpha>1 also requires eigendecomposition on the correlation matrix F, so overall alpha=1 is recommended that enjoys good performance and also avoids expensive eigendecomposition.
>
> “Why the coming back of nuisance features related to the downstream performance?”:
>
> After we learn the invariant features, we also want to pick up the nuisance features because they can capture more details of the inputs and are also useful for downstream performance. But if we learn the nuisance features at the beginning, they can compete with the invariant features with low-level supportive features and lead to worse performance. See more discussions in Section 6. We believe understanding these phenomena require analysis of the non-linear networks, and we leave it as future work.
>
> We hope our response clarifies your concerns and you will consider raising your score. We also thank you for your suggestions in the writing, we have revised our paper accordingly.

---

### Official Review · Reviewer_5xNt · 2021-11-03

**Correctness:** 4
**Technical Novelty And Significance:** 2
**Empirical Novelty And Significance:** 2
**Recommendation:** 5
**Confidence:** 3

**Main Review:**

Positives:
- The work highlights the important role weight decay plays in learning a good representation.
- The work replaces an existing algorithm, DirectPred, with DirectSet and DirectCopy, which has comparable performance while being cheaper and more efficient.
- This work has strong theoretical proofs that support their arguments.

Whereas the proofs are sound, I struggle with appreciating the intended novelty of this work.  The work performed in this paper seems heavily preoccupied with two parts: 1) analyzing DirectCopy, which seems to be a cousin of a previously proposed algorithm DirectPred, as well as the DirectSet family as a whole, and 2) using DirectCopy to show the importance of weight decay.

The analysis of DirectSet and DirectCopy succeeds at proving that it can successfully learn a projection matrix onto an invariant feature space subspace, but essentially boils down to a similar approach as DirectPred (albeit more efficient).  Regarding weight decay, explaining its role, as well as that of other hyperparameters, and why they are important for ncSSL has already been previously explored in [1], and specifically for the BYOL model in prior work.

[1] Tian et al., Understanding Self-Supervised Learning Dynamics without Contrastive Pairs, 2019.

**Summary Of The Paper:**

In this paper, the authors make theoretical progress on understanding non-contrastive self-supervised learning (ncSSL).  ncSSL has previously demonstrated strong empirical performance, even outperforming contrastive learning, but the theory behind it is still unclear.  In this work, the authors build off of prior analysis by [1], and showcase the role weight decay has in learning a desirable representation; it acts as a threshold that discards noisy features with high variance introduced by the data augmentation, and keeps stable features with low variance.

[1] Tian et al., Understanding Self-Supervised Learning Dynamics without Contrastive Pairs, 2019.

**Summary Of The Review:**

Overall, I appreciate the incremental theoretical analysis the authors supply to try to understand non-contrastive self-supervised techniques.  I believe their work is sound and thorough, but would welcome more novel insights in this direction.  I would therefore recommend a weak rejection.

---

> ### Author Response · Authors · 2021-11-16
> **Response**
>
> Thanks a lot for your reviews! We are glad to take this opportunity to address your concerns as below.
>
> "Role of weight decay has been explored in Tian et al 2021":
>
> Tian et al only explained why the representation in nc-SSL does not collapse, but did not study what representation is learned and how the representation is related to the data distribution and augmentation process. In particular, they assumed the augmentation is isotropic in all directions and did not differentiate the invariant features and nuisance features. In our model, we relaxed the isotropic assumption and allow the augmentation to act only in the nuisance subspace. Our analysis shows that weight decay can discard the nuisance features and keep the invariant features, which was not explained in Tian et al.
>
> We hope our response clarifies your concerns and you will consider raising your score. Thank you!

---

### Official Review · Reviewer_6aGh · 2021-11-05

**Correctness:** 4
**Technical Novelty And Significance:** 3
**Empirical Novelty And Significance:** 3
**Recommendation:** 6
**Confidence:** 3

**Main Review:**

\textbf{Originality & Novelty}: The motivation of the paper is to understand why non-contrastive SSL can learn meaningful representations. The paper does provide a new view point on this question by highlighting the importance of the weight decay coefficient. So far as I know, the analyses and the algorithm is novel. At least the paper is a reasonable improvement over DirectPred.

\textbf{Strength}: (1) The paper argues that the weight decay coefficient is the key factor that makes ncSSL learns meaningful representations. This argument is very interesting and deserves more investigation in the future. (2) The paper carefully examines nearly all aspects of DirectCopy with experiments.

\textbf{Weakness}: I think the paper is overall in good shape, with a few minor points: (1) In Fig. 2, even when weight decay is 0, the curves show that some of the eigenvalues are very small after training, and there still exists a 'drop' in the figure, though not as sharp as when weight decay > 0. I wonder why this happens. (2) Basically the analyses are based on SGD. I wonder whether the phenomena will change if other optimizers, like Adam, are used. (3) Another interesting empirical analysis overlooked by the authors is evaluation on more down-stream tasks, like object detection.

**Summary Of The Paper:**

The paper provides a new method for self-supervised learning (SSL), called DirectCopy, based on a previous work DirectPred. An important contribution of this work is theoretical analyses on DirectSet($\alpha$), a theoretical model on linear neural networks that work with arbitrary $\alpha$. The authors prove that the weight decay coefficient $\eta$ has the ability to filter out unnecessary features and hence helps learn useful representations. DirectCopy is a special case of DirectSet($\alpha$) when $\alpha=1$. The new algorithm does not require to compute the burdensome eigen-decomposition while enjoying similar empirical performance as DirectPred.

**Summary Of The Review:**

The paper is solid in both theory and experiments. The argument of this paper that the weight decay coefficient is responsible for learning powerful representations, is valuable and intriguing. To summarize, I recommend acceptance of the paper.

---

> ### Author Response · Authors · 2021-11-16
> **Response**
>
> Thank you so much for your positive review and valuable suggestions! We are glad to answer your questions below.
>
>
> “Eigenvalues of F still drop when weight decay equals zero”:
>
> Even without weight decay, features along different eigen-directions of F can have very different magnitudes, so when the eigenvalues are sorted, there will be a natural drop in the eigenvalues. We focus on the comparison between the eigenvalue spectrum across different weight decays. Larger weight decay leads to less number of large eigenvalues, which suggests that larger weight decay can discard the nuisance features more effectively.
>
>
> “Phenomenon still holds for other optimizers as Adam?”:
>
> Empirically, we observed that DirectCopy still outperforms DirectPred on CIFAR-10 when Adam is used. The analysis on Adam will be more challenging and we leave it as future work.
>
>
> “Evaluation on more down-stream tasks, like object detection”:
>
> Thanks for your suggestion, we will include evaluations on more downstream tasks in a later version.

---

> > ### Comment · Reviewer_6aGh · 2021-12-08
> > **Thanks**
> >
> > Thank you for the response! I'll keep my rating after reading the response.

---

### Author Response · Authors · 2021-11-16
**Differences between our paper and Tian et al 2021**

A common concern between Reviewer 5xNt and Reviewer G6yG is that our algorithm and analysis are based on Tian et al 2021 and some results may seem similar. Here, we clarify the major differences between our paper and Tian et al 2021:

Tian et al only explained why the representation in non-contrastive self-supervised learning (nc-SSL) does not collapse to zero, but did not study what representation is learned and how the representation is related to the data distribution and augmentation process. In particular, they assumed the augmentation is isotropic in all dimensions and did not define the invariant features and nuisance features. In our model, we relaxed the isotropic assumption and allow the augmentation to act only in the nuisance subspace. Our analysis explained how the representation is learned in nc-SSL: weight decay discards the nuisance features and keeps the invariant features.

Motivated by the analysis, we also designed a simpler and more efficient algorithm (DirectCopy), which achieved comparable or even better performances than the original DirectPred across different datasets.

Overall, we view our work as an initial step towards demystifying the representation learning mechanism in nc-SSL. Analyzing the non-convex gradient descent dynamics in nc-SSL is very challenging and there have been very few works before us. Although our theory is restricted to simple linear networks, the analysis is already non-trivial and sheds light on the crucial role of weight decay and leads to more efficient algorithms. We hope this work can inspire more research to further explain the representation learning in nc-SSL.


[1] Tian et al 2021. Understanding Self-supervised Learning Dynamics without Contrastive Pairs.

---

### Decision · Program_Chairs · 2022-01-20

**Decision:**

Reject

**Comment:**

This paper furthers recent work by Tian et al. 2021 to explain how representation learning with non-contrastive self-supervision works. The paper accomplishes this by analyzing a family of algorithms in which DirectPred from Tian et al. (2021) is a special case. Their theoretical analysis is performed with linear networks. Overall, the reviewers questioned the added value relative to Tian et al. 2021, noting that

"The analysis of DirectSet and DirectCopy succeeds at proving that it can successfully learn a projection matrix onto an invariant feature space subspace, but essentially boils down to a similar approach as DirectPred (albeit more efficient)"

The authors in their reply state "how the representation is related to the data distribution and augmentation process," however the relative contribution and why its important isn't transparent.